

# Leveraging TROPOMI observations and WRF-GHG modeling to improve methane emission assessments in India

Thara Anna Mathew[1,2], Dhanyalekshmi Pillai[1,2], Jithin Sukumaran[1,2], Monish Vijay Deshpande[1,2,3], Michael Buchwitz[4], Oliver Schneising[4], Vishnu Thilakan[1,2,5], Aparnna Ravi[1,2], Sanjid Backer Kanakkassery[1,2,6], Sivarajan Sijikumar[7], Imran A Girach[8], and S Suresh Babu[7]

[1]Indian Institute of Science Education and Research Bhopal (IISERB), India
[2]Max Planck Partner Group (IISERB), Max Planck Society, Munich, Germany
[3]Now at the University of Michigan, Ann Harbor, Michigan, USA
[4]Institute of Environmental Physics (IUP), University of Bremen FB1, Bremen, Germany
[5]Now at Lund University, Lund, Sweden
[6]Now at Max Planck Institute for Biogeochemistry (MPI-BGC), Jena, Germany
[7]Space Physics Laboratory (SPL), Vikram Sarabhai Space Centre, Thiruvananthapuram, India
[8]Space Applications Centre (SAC), Indian Space Research Organisation, Ahmedabad, India

**Correspondence:** Dhanyalekshmi Pillai (dhanya@iiserb.ac.in)

**Abstract.**

Atmospheric methane ($CH_4$) contributes to global warming and climate change. Multiple factors control its atmospheric growth rate, posing challenges for climate change mitigation in regions with limited observations, like India. In this study, we examine the potential of dry air column methane mixing ratio ($XCH_4$) observations from the TROPOspheric Monitor-
ing Instrument (TROPOMI) in conjunction with the high-resolution Weather Research Forecast model with Greenhouse Gas module (WRF-GHG) to improve the annual $CH_4$ budget of India. Our analysis demonstrates the potential of WRF-GHG to represent the atmospheric $XCH_4$ and $CH_4$ distributions, including seasonal patterns, albeit with non-negligible uncertainties when compared with satellite and ground-based observations for 2018 and 2019. We find that the WRF-GHG simulations overestimate the $XCH_4$ and underestimate the near-surface $CH_4$ distributions. Our first-order inversion analyses report annual
$CH_4$ emissions ranging from 23.3 to 25.2 Tg with an uncertainty of 3.3 Tg (anthropogenic sources), showing that the current global emission inventories overestimate $CH_4$ emissions considerably. Our estimates are approximately 19% higher than those in the India Fourth Biennial Update Report (19.6 Tg) and close to the latest Global Methane Budget 2000-2020 (21.7 Tg). Overall, this study demonstrates the usefulness of TROPOMI observations for assessing Indian $CH_4$ emissions and shows a way to improve our understanding of how regional processes can modulate atmospheric $CH_4$ mixing ratios. We highlight the
need for expanded observational coverage and an improved carbon assimilation system over India to refine the methane budget in support of global climate goals.





# 1   Introduction

The concentration of atmospheric $CO_2$ has increased by nearly 50% of the pre-industrial levels, while that of $CH_4$ has increased by 150% (Ciais et al., 2014; Myhre et al., 2013a). $CH_4$ is the most prevalent non-$CO_2$ greenhouse gas, with a warming potential 28 times that of $CO_2$ over 100 years and 84 times over 20 years. (Lee et al., 2023; Montzka et al., 2011; Saunois et al., 2016; Stocker et al., 2013) and an atmospheric lifetime of $9.1 \pm 0.9$ years (Prather et al., 2012). Starting from 2007, the concentration of $CH_4$ has proliferated from an annual global mean of 1775 ppb to 1921 ppb in 2024 with a total rise of 146 ppb, which denotes a huge overall growth since the start of industrialisation. The warming of wetlands, an increase in the ruminant population, and a decline in biomass burning—previously masking the rise in isotopically negative fuel use—are some of the key factors that may have contributed to the recent surge in $CH_4$ concentrations (Nisbet et al., 2019). The observed decline in carbon isotope ratio ($\delta^{13}CH_4$) indicates a shift toward increasing biogenic $CH_4$ sources, such as microbial emissions from wetlands and agriculture (has more negative $\delta^{13}CH_4$ signature) rather than fossil fuel or biomass burning contributions (Skeie et al., 2023; Schaefer et al., 2016). Additionally, the decline of hydroxyl radical ($OH^-$), the most dominant $CH_4$ sink, adds to the total atmospheric $CH_4$ budget (Stevenson et al., 2020).

The global stock-take under Article 14 of the Paris Agreement implies the responsibility of each party to prepare, communicate, and maintain the successive nationally determined contributions (NDCs) to climate action (EC 2014). A 30% global reduction in $CH_4$ emissions from 2020 to 2030 has been aimed at the Global Methane Pledge, launched at a 2021 meeting of the United Nations Framework Convention on Climate Change (UNFCCC). Being one of the significant contributors to greenhouse gas (GHG) emissions, India plays an essential role in the global GHG scenario. Still, it lacks sufficient long-term, continuous, and accurate observations of the GHG to quantify the sources and sinks (Zhang et al., 2014). The country holds high $CH_4$ emission potential with its largest cattle population, intense flood irrigation practices, ever-increasing fuel demand, and large wetland extent (nearly 4.7% of its total geographical area) (Ganesan et al., 2017; Garg et al., 2011; Ministry of Environment and Change, 2015; Myhre et al., 2013b). $CH_4$ emissions from enteric fermentation account for about 8% of the total GHG emissions of India's National GHG inventory 2020 (MoEFCC, 2024). Emissions Database for Global Atmospheric Research (EDGAR) inventory provides a global sector-wise emission estimate for $CH_4$. However, the bottom-up approach of EDGAR inventory is limited by its accuracy and temporal resolution owing to uncertainties in the data used and methodologies (e.g. uncertain emission factors, aggregation or interpolation errors and sector distribution). Besides, Indian wetland emissions also show inconsistency in estimations ($\sim$ 5 to 9%) depending on the wetland model used (Bloom et al., 2017). Janardanan et al. (2024) reported large uncertainty in wetland emission inventory data over the Indian domain based on satellite observations



and models. Further, insufficient coverage of highly precise and accurate ground-based observations of CH$_4$ and inadequate access to emission reporting over the country can lead to misrepresentations in global emission inventories.

Atmospheric concentration measurements contain integrated information on the underlying source-sink distribution. Therefore, integrating atmospheric mixing ratio measurements, flux information from bottom-up approaches, and transport model simulations can potentially enhance CH$_4$ estimates through inverse modeling (Jones et al., 2021; Cusworth et al., 2022; Berga-

maschi et al., 2018) and independently evaluate reported flux estimates. Due to sparse ground data and inadequate modeling systems, limited studies have used atmospheric CH$_4$ observations to inform about CH$_4$ emission flux estimates across India. There is an imminent call for measurement that can sufficiently constrain regional emissions in modeling systems (Patra et al., 2016). Recent technological advancements in satellite remote-sensing enable high-resolution-high-density observations to be utilised for this inverse-based quantification when modeling techniques are adequately advanced (Myhre et al., 2013a; Jacob

et al., 2016; Alexe et al., 2015; Buchwitz et al., 2017; Liang et al., 2023; Lu et al., 2022). Ganesan et al. (2017) used a top-down approach to estimate India's CH$_4$ emission for 2010-2015. The above study used column-averaged observations of CH$_4$ from the Greenhouse Gases Observing Satellite (GOSAT) along with aircraft observations from Civil Aircraft for the Regular Investigation of the Atmosphere Based on an Instrument Container (CARIBIC) and a few surface measurements from Indian sites to calculate methane emissions by atmospheric inverse modeling. Since 2009, GOSAT has measured the atmospheric column

for CH$_4$ every three days at a 10 km diameter circle (Butz et al., 2011). Despite some limitations in temporal coverage, the spatial resolution of GOSAT observations is much better than its predecessor, the Scanning Imaging Absorption Spectrometer for Atmospheric Chartography (SCIAMACHY), which the research community has widely used (Butz et al., 2011; Yokota et al., 2009; Turner et al., 2015; Buchwitz et al., 2005; Schneising et al., 2011).

Since November 2017, the more advanced TROPOspheric Monitoring Instrument (TROPOMI) on board the Copernicus

Sentinel-5 Precursor satellite provides much higher-density CH$_4$ observations at a high spatial resolution of 7 $\times$ 7 km$^2$, upgraded to 5.5 $\times$ 7 km$^2$ in August 2019 (Hu et al., 2018; Schneising et al., 2023). TROPOMI measures CH$_4$ at the 2.3 $\mu$m band, with a swath width of 2600 km (Jacob et al., 2016; Cusworth et al., 2018). These observations are expected to capture seasonal fluctuations, which, in turn, will give better insight into the source-sink characteristics and quantification. Hence, in the present study, we explore the potential of TROPOMI measurements in representing the distribution of CH$_4$ fluxes over the

Indian region alongside a spatio-seasonal analysis of the CH$_4$ bottom-up inventory information. We demonstrate the advantage of a modeling system, the Weather Research Forecast model coupled with the Chemistry and Greenhouse Gas module (WRF-GHG), operating at a high resolution comparable to TROPOMI, which may minimise forward model-related uncertainties in the carbon assimilation system over India. The assessment of the forward model, WRF-GHG, in the atmospheric



boundary layer is performed by comparing the atmospheric $CH_4$ simulations with atmospheric measurements from a ground-based site. Finally, the annual $CH_4$ emission estimate is also derived for the period 2018-2019 by incorporating the TROPOMI measurements and WRF-GHG forward model in a first-order atmospheric inversion algorithm.

The paper is organised as follows: Section 2 describes the data and methods used for the study. Section 3 presents data post-processing and inverse analysis, and Section 4 discusses the results of the study from the data analysis conducted. The conclusions of the study are presented in Section 5.

## 2  Data and methods

In this section, we describe the measurements and techniques used for exploring the potential of TROPOMI measurements and the WRF-GHG atmospheric transport model in inferring $CH_4$ distribution over India. A first-order inverse method has been devised to deduce the $CH_4$ fluxes over the Indian region by minimizing mismatches between TROPOMI $CH_4$ measurements and WRF-GHG mixing ratio simulations, thereby correcting the distribution of prior fluxes. Figure 1 shows the model domain with outlines of each geographical region (more details are given in the subsequent sections) considered in this study.

### 2.1  TROPOMI observations

The potential of TROPOMI Sentinel-5p to detect significant sources in the single-pass has been demonstrated in recent publications (Hu et al., 2018; de Gouw et al., 2020; Schneising et al., 2020; Chen et al., 2022; Jacob et al., 2022). We utilized TROPOMI $CH_4$ observations obtained through the Short Wave Infrared (SWIR) band, centered at approximately 2.3 $\mu$m. The atmospheric column-averaged $CH_4$ mixing ratio ($XCH_4$) is retrieved using the Weighting Function Modified Differential Optical Absorption Spectroscopy (WFM-DOAS) algorithm. The WFMD algorithm uses a least-squares approach based on scaling prior atmospheric vertical profiles to retrieve $XCH_4$ and $XCO$ simultaneously (Buchwitz et al., 2007; Schneising et al., 2011). Here, we use the WFMD v1.8 algorithm, for which the efficiency has been validated using Total Carbon Column Observing Network (TCCON) measurements, resulting in an improved random error (12.4 ppb) compared to the previous versions (v1.5 and v1.2) (Schneising et al., 2023). For the analysis, we filtered flagged data and utilized only good-quality retrievals represented by *xch4_quality_flag* = 0.

### 2.2  Modeling system for atmospheric $XCH_4$ mixing ratio simulations

We have used the Weather Research and Forecasting model coupled with the Chemistry and Greenhouse Gas module (hereafter referred to as WRF-GHG) for atmospheric $CH_4$ transport simulations. The core component is the WRF model, based on



fully compressible, non-hydrostatic Eulerian equations on terrain-following vertical grids for simulating atmospheric transport (Skamarock et al., 2008). The GHG-TRACER package allows the online passive tracer transport of $CH_4$ mixing ratio in the atmosphere (Beck, 2011; Pillai et al., 2016). The input fluxes from each emission sector are separately provided as "tagged" tracers when added to the first layer in the modeling system. This allows the decoupling of emission contributions to the total atmospheric $CH_4$ mixing ratio. We have used the WRF-GHG 3.9.1.1 version with a horizontal resolution of $10 \times 10$ km$^2$

(Lambert conformal conic projection grid) and a temporal resolution of 1 hour. The model covers the Indian domain with 307 $\times$ 407 grid points and 39 vertical levels. The fifth generation ECMWF reanalysis (ERA-5) data with a horizontal resolution of $0.25° \times 0.25°$ and temporal resolution of 6 hours with 137 vertical levels are used as initial and boundary conditions for meteorology. The model is re-initialized each day with ERA5 meteorology, in which a 6-hour spin-up time was configured. For $CH_4$ mixing ratio fields, initial and boundary conditions are prescribed from the Copernicus Atmosphere Monitoring

Service (CAMS re-analysis data). CAMS provides the simulated atmospheric mixing ratios of $CH_4$, with a spatial resolution of $0.25° \times 0.25°$ and a temporal resolution of 6 hours on 60 vertical levels. The model utilizes these initial fields to represent the far-field flux contribution (background) to the total mixing ratios. Similar to emission flux contributions, we disentangled the background contribution in the model output to investigate its impacts separately. We have also considered the reported level of uncertainties in CAMS-simulated mixing ratios (e.g. Agustí-Panareda et al. (2023); Wang et al. (2023)) and applied a

monthly factor correction (1 to 3% correction for the whole model domain) to our CAMS-simulated WRF-GHG background to minimize the unrealistic representation of far-field contribution to XCH$_4$ levels. The corrected WRF-GHG background mixing ratios are hereafter termed simply as background mixing ratios.

    We used the Emission Database for Global Atmospheric Research (EDGAR v7.0; Crippa et al. (2024)), Global Fire Assimilation System (GFAS v1.2; Kaiser et al. (2012)), and a global wetland $CH_4$ emissions and uncertainty dataset for atmospheric

chemical transport models (WetCHARTs 1.3.1; Bloom et al. (2021)) as prior emission fluxes to represent anthropogenic, biomass burning and wetland emissions respectively. We applied time factors to the annual EDGAR dataset based on step-function time profiles (Kretschmer et al., 2014) and converted it to 1-hour temporal resolution. GFAS data with $0.1° \times 0.1°$ resolution represents biomass burning emissions in the model. GFAS emissions are calculated using fire radiative power observations from the Moderate Resolution Imaging Spectroradiometer (MODIS) instrument aboard the Terra satellite.

WetCHARTs is an ensemble dataset that provides gridded emissions data from 2001 to 2019 at a resolution of $0.50° \times 0.50°$ (Bloom et al., 2017), which were then re-gridded to $0.1° \times 0.1°$ with a temporal resolution of 1 hour. The simulated total atmospheric $CH_4$ mixing ratio thus contains contributions from the initial fields as well as anthropogenic, biomass burning,



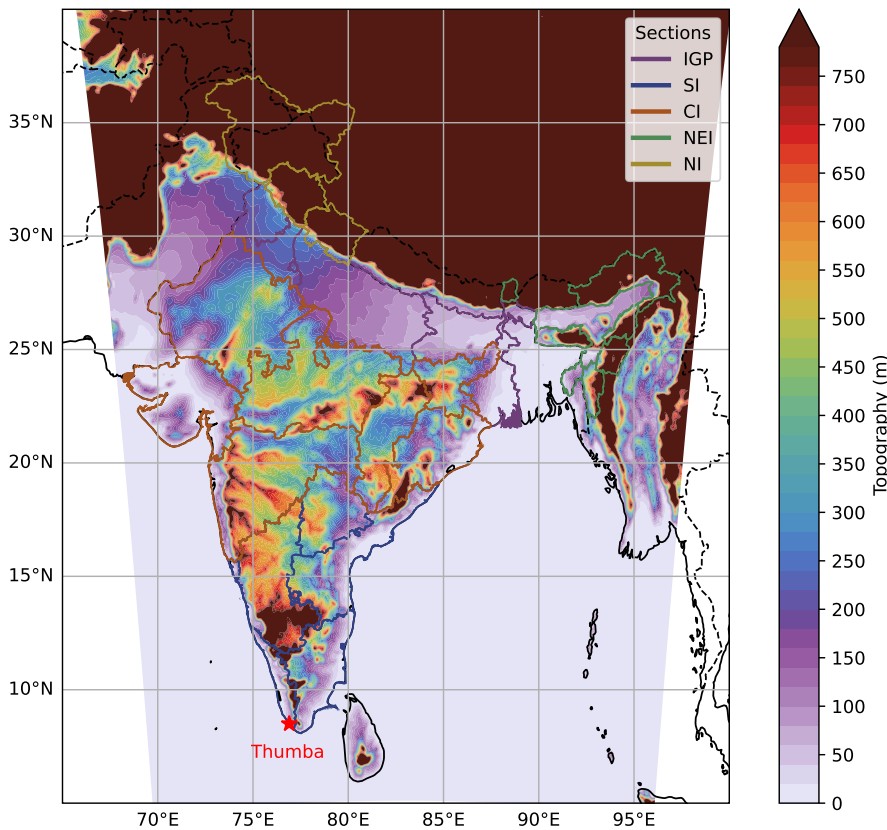

**Figure 1.** The topographical height contour of the model domain overlaid with outlines of each geographical section of the Indian landmass according to Survey of India (2024). CI stands for Central India, NEI for North East India, NI for North India, SI for South India, and IGP for the Indo Gangetic Plain regions.

and biogenic fields of CH$_4$. Table 1 summarizes the WRF-GHG model set-up used in this study. The general meteorological configuration for the WRF-GHG model set-up applied here is described in Thilakan et al. (2022).

**2.3 Ground-level observations**

To assess the model's performance at the surface level, a comparative analysis was conducted using CH$_4$ *in situ* measurements from a ground-level pollution monitoring station in Thumba (8.5°N, 76.9°E) as denoted in Fig. 1 for 2018 & 2019. Located in southwestern India, Thumba is a tropical coastal station approximately 10 km northwest of Thiruvananthapuram and 500 m inland from the Arabian Sea. CH$_4$ concentrations were measured using a greenhouse gas analyzer (model: 911–0011-1001)

by Los Gatos Research, USA, based on the off-axis integrated cavity output spectroscopy (OA-ICOS) method (Baer et al., 2002; Raju et al., 2022; Sijikumar et al., 2023; Uma et al., 2024). Air samples were collected from about 10 m above ground

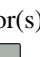
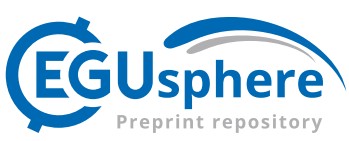

**Table 1.** WRF-GHG model configuration.

| Parameter | Details |
|---|---|
| Domain Configuration | Single domain with a horizontal resolution of 10 km; 39 vertical levels; 307 × 407 grid points |
| Vertical coordinates | Terrain-following hydrostatic pressure vertical coordinates |
| Basic equations | Non-hydrostatic; compressible |
| Grid type | Arakawa C grid |
| Time integration | Third-order Runge–Kutta split explicit |
| Spatial integration | Third- and fifth-order differencing for vertical and horizontal advection, respectively; both for momentum and scalars |
| Time step | 60 s |
| Physics schemes | |
| Radiation | Rapid Radiative Transfer Model (RRTM) for longwave and Dudhia for shortwave |
| Microphysics | WRF single-moment three-class (WSM3) classic simple ice scheme |
| PBL | Yonsei University (YSU) scheme |
| Surface layer | Monin–Obukhov |
| Land surface | NOAH land surface model (LSM) |
| Cumulus | Grell–Freitas ensemble scheme |
| Emission fields | |
| Flux type | Product, Version, Spatial resolution, Temporal resolution, Source/website, Reference |
| Anthropogenic | EDGAR, v7.0, 10 km, Annual, https://edgar.jrc.ec.europa.eu/, (Crippa et al., 2024) |
| Biomass burning | GFAS, v1.2, 10 km, Daily, http://apps.ecmwf.int/datasets/data/cams-gfas/, (Kaiser et al., 2012) |
| Biospheric | WetCHARTs V.1.3.1, 0.5 °, Monthly, https://daac.ornl.gov/cgi-bin/dsviewer.pl?ds_id=1915, (Bloom et al., 2017) |
| Initial and lateral boundary conditions | |
| Field | Product, Version, Spatial resolution, Temporal resolution, Source/website, Reference |
| Meteorology | ERA5, NA, 25 km, 1 h, https://cds.climate.copernicus.eu/cdsapp#!/home,(Hersbach et al., 2020) |
| Tracer | ECMWF/CAMS, gqiq, 50 km, 6 h, http://atmosphere.copernicus.eu,(Agustí-Panareda et al., 2023) |



level (AGL) using the analyzer's internal pump. Calibration was performed periodically using $CH_4$ standards supplied by the National Oceanic and Atmospheric Administration (NOAA), USA. Measurements were recorded at 1-second intervals, with hourly averages used for subsequent analysis. $CH_4$ measurement uncertainty is 0.25% (i.e. 5 ppb with respect to 2000 ppb of

$CH_4$) and the reported precision ($1\ \sigma$) is 1 ppb.

## 3    Data post-processing and inverse analysis

We regridded the daily total dry column mixing ratio of $CH_4$ from TROPOMI at $0.10° \times 0.10°$ resolution, covering a period of 2018 to 2019. From the hourly WRF-GHG $CH_4$ mixing ratio simulations generated, we sampled those corresponding to the TROPOMI overpass time for the model domain. We applied the satellite's averaging kernel (AK), as shown in Equation 1,

for the model evaluation with observations by considering the vertical sensitivity of the satellite instrument (Schneising et al., 2019). AK is proportional to the sensitivity profile of the measurement that is weighted with the assumed tracer profiles and provides the relation between the retrieved and known tracer profiles. i.e. Applying satellite AK to the model simulations at different vertical levels minimizes the mismatches owing to the instrument vertical sensitivities to the column observations (Eskes and Boersma, 2003; Wang et al., 2023; Schneising, 2024).

We applied the AK to the modeled dry-air $CH_4$ profiles and derived the dry-air column-averaged mixing ratio of $CH_4$, $\Upsilon_{mod}$, as follows:

$$\Upsilon_{mod} = \sum_l \left( \Upsilon_{apr}^l + A_l \left( \Upsilon_{mod}^l - \Upsilon_{apr}^l \right) \right) w_l \tag{1}$$

where $l$ is the index of the vertical layer, $A_l$ is the averaging kernel and $\Upsilon_{apr}^l$ is the *a priori* mole fraction of layer $l$, and $\Upsilon_{mod}^l$ is the corresponding simulated mole fraction of layer $l$. $w_l$ is the layer-dependent pressure weight.

Hence, $\Upsilon_{mod}$ (= $XCH_{4,mod}$) is used for the model-observation comparisons and inversion analyses. i.e., in this study, $\Upsilon_{mod}$ represents WRF-GHG $XCH_4$ simulations.

### 3.1    Estimating optimized $CH_4$ flux over India

We performed a simple Bayesian inverse optimization to deduce the improved emission estimates over the Indian domain. The inversion is designed in such a way that it describes the relationship between the mixing ratio observations and the surface

flux emission information (the unknown state) and *a priori* information available. This approach allows us to identify the





class of possible states consistent with the available information and to assign a probability density function (pdf) to them. The quantities to be optimized, represented by the state vector $x$ with $n$ elements $x_1, x_2, x_3, \ldots, x_n$ correspond to the total monthly emission information averaged over each political state in India. Here, $n$ represents 36 state regions of India. The measured quantities, represented by the measurement vector $y$ with $m$ elements $y_1, y_2, y_3, \ldots, y_m$ , represent the total column

observations from TROPOMI over a month at $0.1° \times 0.1°$ spatial resolution. Here, $m$ represents the total number of observations available in each political state.

The relationship between the measurement vector, $y$, and the state vector, $x$, can be written as:

$$y = \mathbf{F}(x) + \epsilon \qquad (2)$$

where $\mathbf{F}(x)$ encapsulates the physics of the measurements as a function of the state vector, described here by our forward

transport model, WRF-GHG. The error term $\epsilon$ includes model error, representation error (sampling mismatch between the observations and the model), and measurement error.

Linearizing the forward model to a reference state yields:

$$\mathbf{F}(x) = \mathbf{K}x + \epsilon \qquad (3)$$

Here, $\mathbf{K}$ is the $m \times n$ Jacobian matrix, representing the sensitivity of the mixing ratio simulated by the forward model to the

state vector. The elements of $\mathbf{K}$ are thus: $k_{m,n} = \frac{\partial \mathbf{F}_m(x)}{\partial x_n}$

Since we have not implemented the adjoint model for our forward transport model, the Jacobian was constructed by applying the transport model (WRF-GHG) to the perturbed emissions over the target region (each political state) and obtaining the sensitivity to the corresponding observations, as follows:

$$\mathbf{K} = \frac{(\mathbf{\Upsilon} + \Delta\mathbf{\Upsilon}) - \mathbf{\Upsilon}}{_{TR}\mathbf{\Phi}_{\text{perturbed}} - _{TR}\mathbf{\Phi}} \qquad (4)$$

Here, $\mathbf{\Upsilon}$ is the column mixing ratio field, and $(\mathbf{\Upsilon} + \Delta\mathbf{\Upsilon})$ is the perturbed column mixing ratio field over the target region. $\mathbf{\Phi}$ is the emission flux field, and $\mathbf{\Phi}_{\text{perturbed}}$ is the perturbed emission flux field over the target region.

In our implementation, we focus on anthropogenic fluxes and their contributions to the atmospheric dry air column mixing ratios. $x_A$ represents the prior fluxes, which consist of the spatially averaged monthly anthropogenic (major contributions from



enteric fermentation, agricultural soil, waste water handling, and fuel exploitation) and biomass emissions, separately for each

political state. The far-field contributions are removed from the observations to optimize the local enhancement fluxes. i.e., the

measurement vector $\boldsymbol{y}$ consists of total column observations from TROPOMI subtracted by background mixing ratios over a

month at $0.1° \times 0.1°$ spatial resolution.

The background mixing ratios are simulated by WRF-GHG, as explained in Sect. 2.2. We have considered measurement

errors (including forward model errors) and prior errors: $\mathbf{S}_e$ and $\mathbf{S}_a$ represent measurement error and prior error covariance

matrices, respectively. The measurement error covariance matrix $\mathbf{S}_e$ consists of $CH_4$ retrieval ($\boldsymbol{y}$) and the forward model ($\mathbf{F}(\boldsymbol{x})$)

errors . We assumed a prior emission uncertainty of 80% and a measurement uncertainty of 16 ppb, as adopted from Liang

et al. (2023), calculated using the residual error method (Heald et al., 2004). We have not considered cross-correlations; hence,

only the diagonal elements of the matrix are non-zero.

The solution to the inverse problem is obtained by minimizing the Bayesian scalar cost function $J(\mathbf{x})$ (Rodgers, 2000):

$$J(\boldsymbol{x}) = (\boldsymbol{x} - \boldsymbol{x}_A)^{\mathrm{T}}\mathbf{S}_a^{-1}(\boldsymbol{x} - \boldsymbol{x}_A) + (\boldsymbol{y} - \mathbf{K}\boldsymbol{x})^{\mathrm{T}}\mathbf{S}_e^{-1}(\boldsymbol{y} - \mathbf{K}\boldsymbol{x}_A) \tag{5}$$

where $\nabla_{\mathbf{x}}J(\mathbf{x}) = 0$, the optimal estimate $\hat{\mathbf{x}}$ is obtained (Rodgers, 2000). The state that maximizes the posterior pdf $P(\mathbf{x}|\mathbf{y})$ is

the maximum a posteriori solution (MAP). The maximum a posteriori solution is obtained as follows:

$$\hat{\boldsymbol{x}} = \boldsymbol{x}_A + \left(\mathbf{K}^{\mathrm{T}}\mathbf{S}_e^{-1}\mathbf{K} + \mathbf{S}_a^{-1}\right)^{-1}\mathbf{K}^{\mathrm{T}}\mathbf{S}_e^{-1}\left(\boldsymbol{y} - \mathbf{K}\boldsymbol{x}_A\right) \tag{6}$$

Thus, $\hat{\boldsymbol{x}}$ represents the optimized spatially averaged monthly anthropogenic and biomass burning fluxes corresponding to

each political state considered. The posterior error covariance matrix, denoted by $\hat{\mathbf{S}}$, is derived as follows.

$$\hat{\mathbf{S}} = (\mathbf{K}^{\mathrm{T}}\mathbf{S}_e^{-1}\mathbf{K} + \mathbf{S}_a^{-1})^{-1} \tag{7}$$

The error reduction for each month following the inversion procedure ($\tilde{e}$) is calculated as follows:

$$\tilde{e} = 1 - \frac{\sigma_{post}}{\sigma_{prior}} \tag{8}$$

where $\sigma_{post}$ is derived as the square root of the diagonal elements of $\hat{\mathbf{S}}$. Similarly, $\sigma_{prior}$ is obtained from the square root of

the diagonal elements of the prior error covariance matrix.





The national budget for annual optimized fluxes $\hat{\boldsymbol{x}}_{annual}$ is derived as:

$$\hat{\boldsymbol{x}}_{annual} = \sum_{t=1}^{N_t} \sum_{s=1}^{N_s} \hat{\boldsymbol{x}}_{s,t}, \tag{9}$$

where $\hat{\boldsymbol{x}}_{s,t}$ represents the estimated value at each political state $s$ for the corresponding month $t$. $N_s$ is the total number of political states, and $N_t$ is the total number of months considered.

## 4   Results and Discussion

### 4.1   Regional distribution of CH$_4$ sources

In this section, we present a comparative assessment of different CH$_4$ sources based on bottom-up inventories such as EDGAR v8.0 (latest release; Crippa et al. (2024)), WetCHARTs v1.3.1, and GFAS v1.2. Though with inherent uncertainties, these bottom-up inventories relate the emissions to the causative processes by considering the emission activities and emission factors, thereby providing us with a "first guess" to identify the prominent sources (Miller and Michalak, 2017). Major CH$_4$ emission sources, such as enteric fermentation, wastewater handling, rice agricultural land, wetlands, and biomass burning, are included in detail to assess the sectoral and regional distributions.

Our sector-wise analysis of the bottom-up inventories shows that the enteric fermentation associated with the digestive process in cattle makes a significant contribution to CH$_4$ emissions in India (42.9 %), followed by wastewater treatment (19.2 %), agricultural soil (12.4 %), fuel exploitation (6.7 %), and the wetland (5.2 %, excluding the agriculture) (see Fig.2). The seasonal sources of CH$_4$ include agriculture (see Fig.2 (b)) and biomass burning, also reported in Ganesan et al. (2017). Figure 2. (c)-(f) shows an annual average of the spatial distribution of CH$_4$ emissions for the four major emission sectors in 2018. The positive anthropogenic trend in CH$_4$ emissions in India can be expected owing to the large cattle population and agricultural activities (especially rice production and waste management) as indicated by 2010-2015 GOSAT observations (Maasakkers et al., 2019). The annual CH$_4$ emissions corresponding to rice cultivation (emission from the sector 'Agricultural soil' as given by the EDGAR inventory) show a peak in values over the Indo-Gangetic Plain (IGP) region (Fig.2 e). Several other studies have used GOSAT and other data sources to analyse the CH$_4$ emissions from rice paddies in India (Miller et al., 2019; Ganesan et al., 2017; Anand et al., 2005). Previous studies reported CH$_4$ emissions from rice paddies in India of about 3.9 Tg yr$^{-1}$, with the bulk emitted between June and September (Commission et al., 2011; Garg et al., 2011; Panigrahy et al., 2010). The sector-wise analysis of the EDGAR inventory for the year 2018 shows significant CH$_4$ highs on the eastern coast, including West Bengal and Odisha, which can be attributed to the large rice cultivation in these regions (Crippa et al., 2023). The analysis



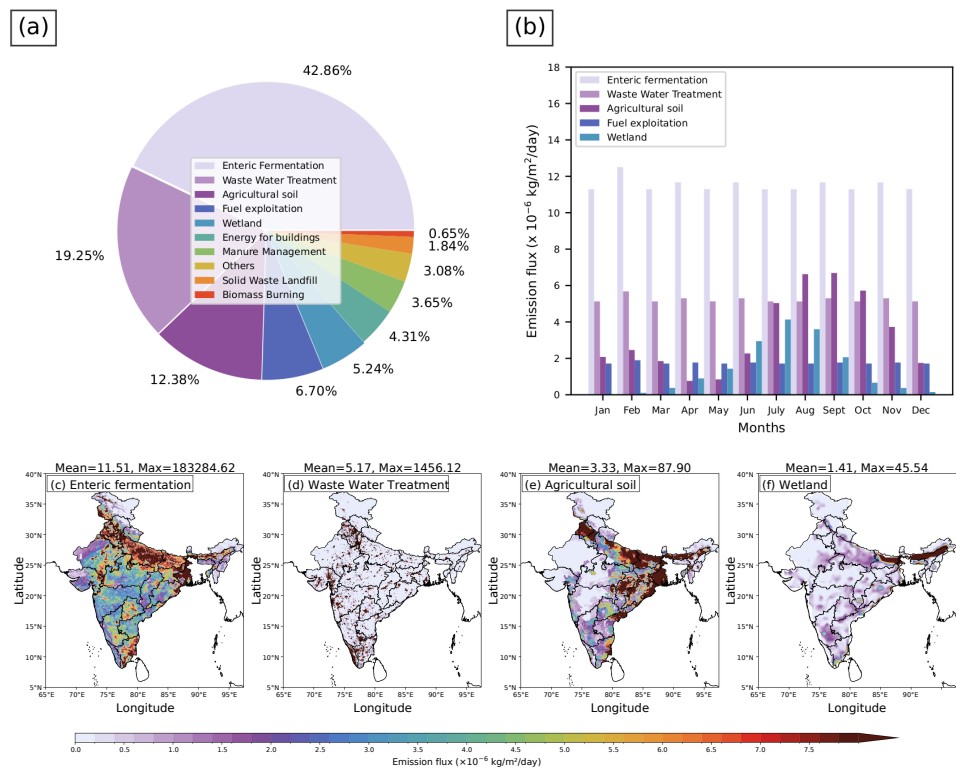

**Figure 2.** (a) The percentage of contributions of different CH$_4$ sources towards total annual emission flux over the Indian domain, (b) monthly contribution (calculated from the EDGAR temporal profiles described in Crippa et al. (2020)) from each source, and (c-f) the spatial distribution of the annual CH$_4$ emission from Enteric fermentation, Waste Water Treatment, Agriculture, and Wetlands, respectively, for 2018.

of monthly emissions from rice cultivation over the year 2018 indicates an increasing pattern in summer monsoon seasons (see "Agricultural soil" in Fig. 2 (b), with the maximum in August (16.9%) and September (16.5%) and the least in April (1.9%). There is a smaller peak in February-March time, owing to winter rice cultivation, which comprises 14% of total rice production

in India (Manjunath et al., 2006). Similarly, in the wetland emissions, we also see an increase in monsoon months (Fig. 2 b). The peak wetland emissions are seen in July (24.6%), followed by August (21.4%), and the least in January (0.2%). It has been previously reported from satellite observations that the waterlogged areas increase nearly threefold from the beginning to the end of the monsoon, resulting in increased wetland CH$_4$ emissions (Agarwal and Garg, 2009). The pre-monsoon CH$_4$ emission (15.5%) is higher than post-monsoon (6.2%; see Fig. 2 (b)). This is due to higher temperatures during the pre-monsoon season

as described in Das et al. (2023). Our analysis of monthly emissions based on GFAS shows a peak due to biomass burning in March (65.4%), with the lowest burning reported in July (0.1%). Other sources of CH$_4$, including fossil-fuel emissions,





**Figure 3.** Seasonal mean emissions from (top panels) wetland, (middle panels) biomass burning, and (bottom panels) anthropogenic sources, separated for seasons, for each region as specified in Figure 1. Note: The ranges of the y-axes are not uniform in panels to improve visibility.





enteric fermentation, and wastewater handling, have not shown considerable seasonal variability. A similar pattern has also been observed from the 2019 anthropogenic and natural $CH_4$ emission analysis (not shown).

Further, anthropogenic emissions continue to be the most significant contributors to the regional $CH_4$ emissions (see Fig. 3 245 & S1). The total anthropogenic emissions (sum of various components like enteric fermentation, agriculture soil, wastewater treatment, and fuel exploitation) reported by the EDGAR bottom-up inventory (version 8) shows a seasonal peak over the SI region ( $> 700 \times 10^{-12}$ kg m$^{-2}$ s$^{-1}$ ), followed by IGP ( $> 400 \times 10^{-12}$ kg m$^{-2}$ s$^{-1}$ ). The least anthropogenic emissions are seen over NI, which is about a factor of 10 less than that over IGP (see Fig. 3). Generally, the seasonal emission peak is seen in October to November, followed by June to September. The emissions are seen at a minimum from March to May. 250 Accordingly, the total anthropogenic emissions over India (sum of all the given regions as seen in Fig. 3 (i)- (l)) indicate a peak in October to November and a dip in March-May (ranging from $\sim$ 1410-1090 $\times$ 10 $^{-12}$ kg m$^{-2}$). While IGP shows consistently high emission distribution spatially (see Fig. 2 (c)-(f)), SI contributes more total emissions than IGP owing to the presence of emission hotspots. We have identified four hotspots in SI: Namakkal (11.25 $°N$, 78.15 $°E$; in Tamil Nadu), Mandapaka Rural (16.75 $°N$, 81.65 $°E$; in Andhra Pradesh), Pasumamla (17.35 $°N$, 78.65 $°E$; in Telangana) and Mulkalappalli, (18.65 255 $°N$, 79.55 $°E$; in Telangana). When those hotspot regions are excluded from SI (labelled as SI$_A$), the highest contribution for total emissions is seen over IGP (see Fig. 3 (i) - (l)). Here, Namakkal in Tamil Nadu has the issue of poultry waste deposition (Ramasamy and Manivel, 2019), possibly generating more volatile gases. Mandapaka, as part of the West Godavari district that is known as "Annapurna" (meaning rice bowl) of Andhra Pradesh, has increased agricultural rice emissions (Gururaj Katti et al., 2002). Pasumamla in Rangareddy district in Telangana has the local poultry industry dumping large amounts of litter 260 in landfills around farms (https://rangareddy.telangana.gov.in/animal-husbandry; last accessed: 12 Feb 2025). Mulkappalli is a coal mine location (https://khammam.telangana.gov.in/economy; last accessed: 12 Feb 2025), contributing to the $CH_4$ high in the inventory data in SI.

Based on WetCHARTs (version 1.3.1), the emissions from natural wetlands are approximated to be 1.7 Tg yr$^{-1}$. The regional wetland emissions are high in June to September ( $\sim$ 185 $\times$ 10 $^{-12}$ kg m$^{-2}$). The wetland emissions show a peak over the SI 265 region i.e. $> 25 \times 10^{-12}$ kg m$^{-2}$ s$^{-1}$ for three seasons (June-September, October-November, March-May) with a peak emission $\sim$ 50 $\times$ 10$^{-12}$ kg m$^{-2}$ s$^{-1}$ (see Fig. 3 (a)-(d)). The magnitude is very low for the other regions except for the June to September season. These seasonal peaks can be associated with the possible increased extent of waterlogged areas in monsoon, as discussed previously, with reference to Fig. 2 (b). The biomass burning emissions, i.e., those of lesser magnitude compared to total anthropogenic emissions and wetland emissions, generally show a peak in March to May in all regions except for the IGP 270 regions. In the IGP region, the burning of Kharif crop residue occurs from October to November, as mentioned by Deshpande





et al. (2022, 2023), which can also be seen in Fig. 3). The maximum seasonal peak is seen in the NEI region during March to May ($\sim 47 \times 10^{-12}$ kg m$^{-2}$), which is about an order higher than that seen over IGP in October-November (see Fig. 3g). This peak in NEI in the March-May season could be due to the extensive slash and burn for clearing weeds preceding the planting season (Deshpande et al., 2023). Total emissions (sum of anthropogenic emissions from EDGAR, wetland emissions from WetCHARTs and Biomass Burning emissions from GFAS) follow regional patterns of anthropogenic emissions and peak dur-
ing October-November and June-September (Fig. S2 (a) & (d)), with the SI region contributing the highest share at 49.2% and 46.4%, respectively. Similarly, SI accounts for the largest contribution in March-May (52%) and December-February (51.7%).

Though the above estimations give an overview of Indian CH$_4$ source contributions and their regional patterns, there have been increased concerns about their accuracy due to methodological weaknesses and data gaps (Solazzo et al., 2021; Madrazo
et al., 2018). For example, the combined emissions from Oil, Gas, and Coal over the Indian region reported by Scarpelli et al. (2025) is 1.8 Tg yr$^{-1}$, whereas EDGAR reported 2.2 Tg yr$^{-1}$. Also, bottom-up methods can overestimate or misinterpret emission sources even at the global level (Saunois et al., 2016). Though inverse modeling can offer better scope in updating the CH$_4$ budget, those approaches are limited by the insufficient coverage of mixing ratio observations and inadequate representation of the process and spatiotemporal distributions in the forward models. Satellite instruments such as TROPOMI may aid in high
observation density with good spatial coverage (Palmer et al., 2021), the potential of which in the Indian context is explored in further sections.

## 4.2 Anthropogenic XCH$_4$ mixing ratio enhancements

In this section, we discuss the mixing ratio enhancements in the atmospheric column in response to spatial and temporal distributions of regional sources for the period 2018-2019. i.e. anthropogenic XCH$_4$ mixing ratio enhancements consider only
contributions from local sources, such as anthropogenic, biomass, and wetland emission sources over the model domain from the emission inventories, not using CAMS-derived background XCH$_4$ (see section 2.2). The IGP and surrounding regions exhibit significant XCH$_4$ enhancements (>60 ppb) from background contribution (far-field fluxes) attributed to anthropogenic and biomass-burning fluxes (see Fig. 4). Seasonally, compared to summer, the highest XCH$_4$ enhancements occur during winter (with a maximum of $\sim 251$ ppb in January). The minimum enhancement is seen in the monsoon time, possibly due to
large-scale mixing in the atmosphere. The increased enhancements in October and November align with additional emissions from agricultural activities such as biomass burning that prevail in some parts of the country. A similar seasonal pattern is also observed in 2019 (see Fig. S3).



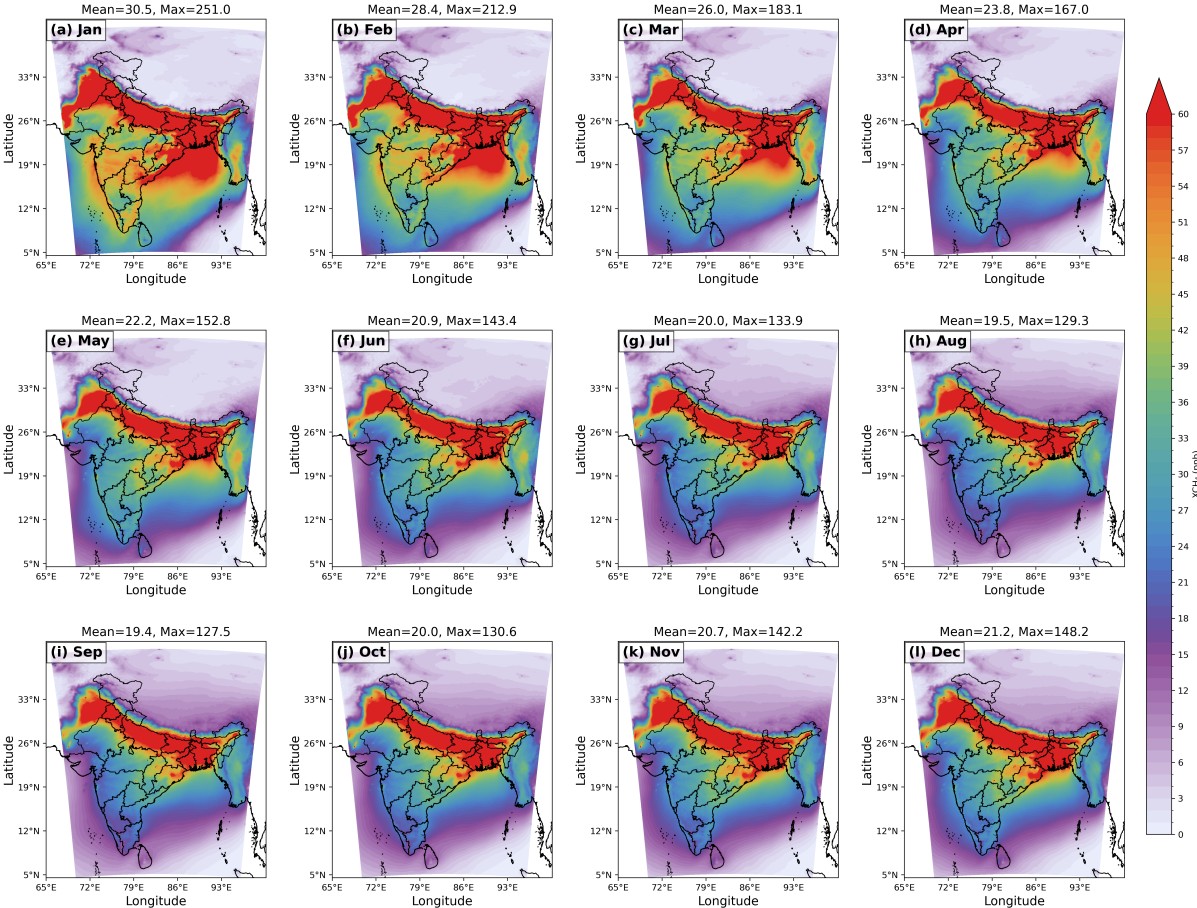

**Figure 4.** Spatial distribution for WRF-GHG simulated anthropogenic mixing ratio enhancement of $XCH_4$ (including biomass burning), separated (a)-(l) for each month of 2018.

Figure 5 (also see Fig. S4) shows the regional variability of anthropogenic $XCH_4$ enhancements across different parts of India as shown in Fig. 1. The highest regional magnitudes and variability in mixing ratio enhancements occur from October to November. Here, consistently high spatial distribution is found over the IGP region (with a median of $\sim 55$ ppb), showing maximum values over the SI region ($\sim 100$ ppb) owing to emission hotspots. During winter, the NEI shows increased $XCH_4$ enhancements, with values reaching up to 115 ppb (with a median value of $\sim 60$ ppb). From June to September, SI enhancements reach up to $\sim 95$ ppb (with a median value of $\sim 45$ ppb). A similar trend is seen for 2019 (Figure not shown). SI exhibits the widest interquartile range with the lowest minimum value, a relatively low median, and the highest maximum in most seasons, indicating the influence of hotspot emissions, as discussed earlier in Section 4.1.



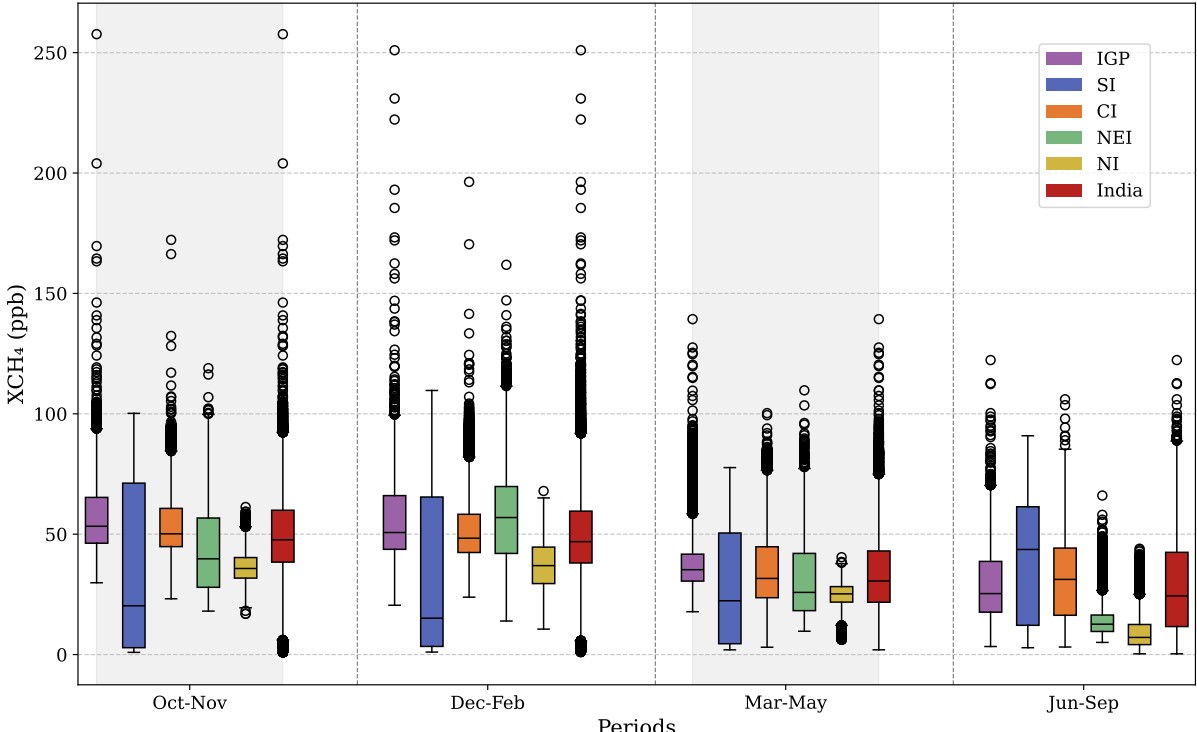

**Figure 5.** Distribution of seasonal average of simulated mixing ratio enhancement ($XCH_{4,ant} + XCH_{4,bbu}$) over different regions of India in 2018. The box plot displays medians, interquartile ranges, and minimum and maximum values, with data points beyond 1.5 times the interquartile range represented as outliers.

## 4.3 Comparison of modeled and observed total $XCH_4$

In this section, we present our comparisons of WRF-GHG simulations with TROPOMI observations of total $XCH_4$ in 2018, considering all months in which reasonable satellite measurements are available after filtering. The details of filtering are provided in Schneising et al. (2023). TROPOMI observations show distinct seasonal variations in the large spatial domain

(Fig. 6), possibly resulting from both atmospheric transport and surface emissions variations. Observations indicate enhanced values in the mean spatial distribution of $XCH_4$ from October to November within the range of $\sim 1862$ to $1870$ ppb. These seasonal increments can be attributed to the shallowing of PBL, which accumulates the effect of increased surface emissions at lower atmospheric levels. These increased local emissions, also seen in Fig. 3, are typical for some parts of India during the October-November season. However, we cannot neglect the likelihood of increased aerosol loads, which impacts $XCH_4$

retrievals (Lorente et al., 2021; Hu et al., 2018; Pandey et al., 2019). All the months show significantly higher total $XCH_4$ mixing ratios over the IGP region in comparison with the other regions over India. The IGP region is more prone to biomass



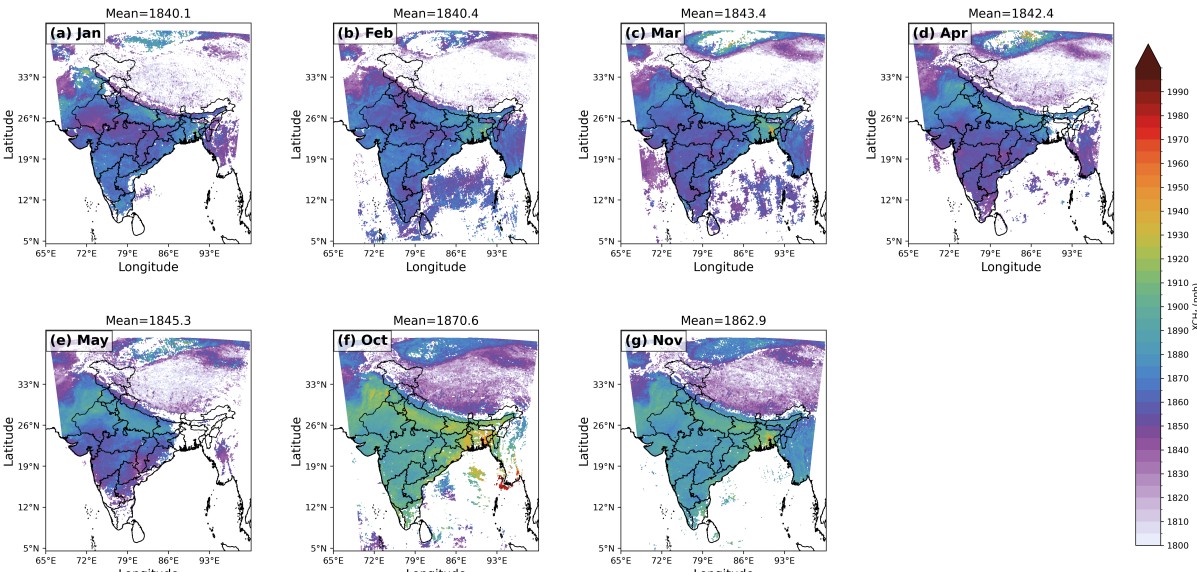

**Figure 6.** Spatial distribution of the TROPOMI Sentinel-5P measurements, averaged for (a)-(g) each available month in 2018. Some months are excluded due to insufficient data points due to filtering using the quality flag as given in the data product.

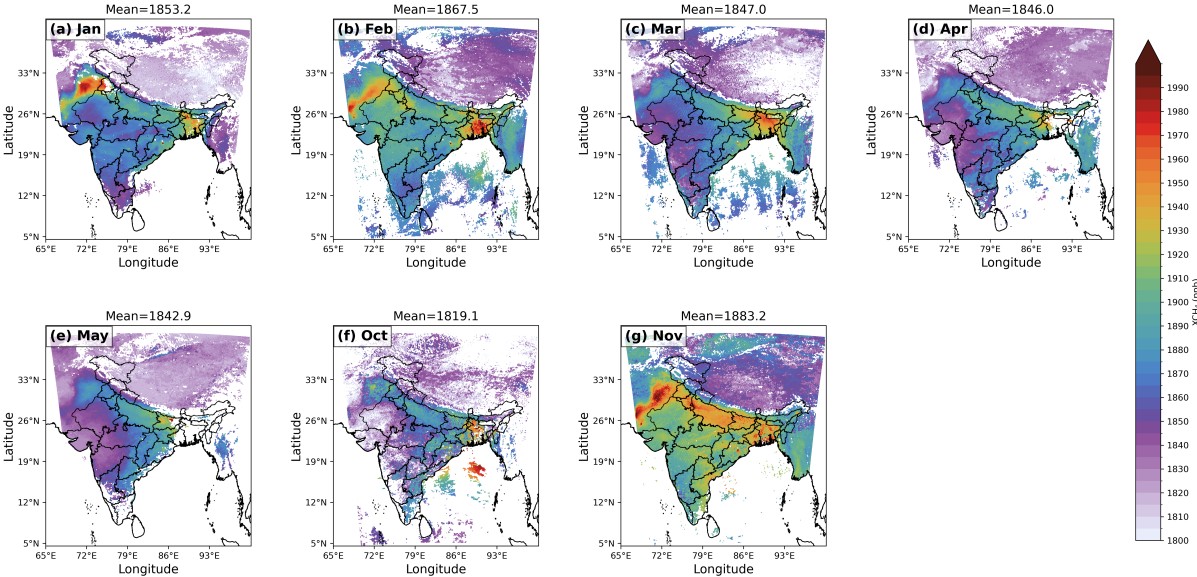

**Figure 7.** Same as Fig. 6, but showing WRF-GHG simulations of total XCH$_4$

burning in October and November, causing more aerosols in the region. We have examined the particulate matter (PM2.5) content using the MERRA database, which indicates a heavier aerosol content due to burning during the winter, not always necessarily peaking in October but during the October-February season (Figure not shown). There is a gradual increase in



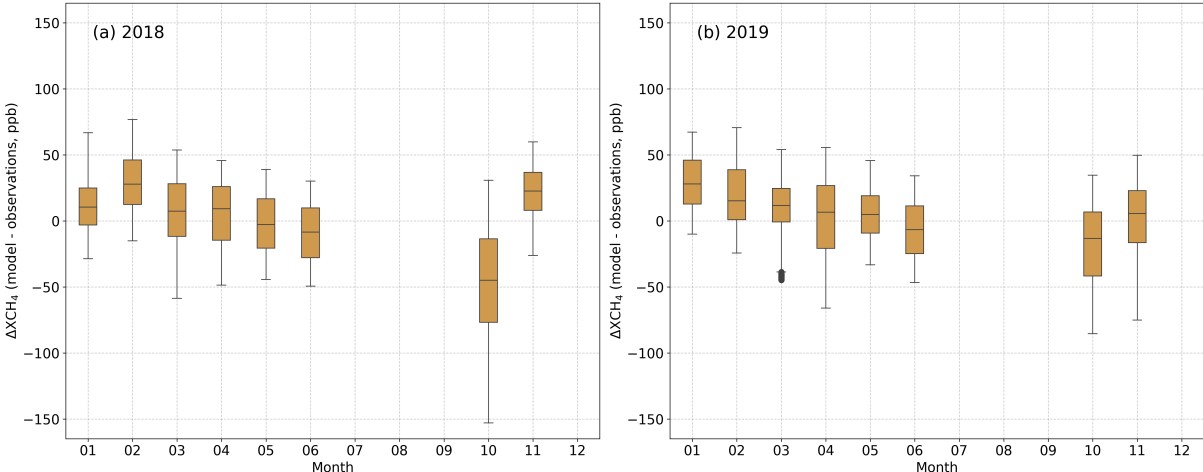

**Figure 8.** Monthly distribution of the difference between WRF-GHG simulations and TROPOMI Sentinel-5p retrievals of $XCH_4$ for each available month in 2018 and 2019 when sufficient observations are available (Outliers are not removed; instead, a 90 % winsorization (Wilcox, 2005) is applied for the outliers.)

$XCH_4$ mixing ratios beginning from the winter month of January till March, with a slight dip in April and then a distinctly high increment in October (exceeding 1870 ppb) in October over the IGP (see Fig. 6. (a)-(f)).

     We find that the WRF simulations generally overestimate the total $XCH_4$ mixing ratio over the Indian region compared to TROPOMI observations, peaking in winter months (>1850 ppb) (see Fig. 7). The IGP emission hotspot is also pronounced in the EDGAR inventory (Fig. 2, see Sect. 4.1), suggesting the increased impact of anthropogenic emissions on the observed

total $XCH_4$. Further, the sectoral analysis of the EDGAR emission inventory and the consistency of the spatial pattern with TROPOMI observations indicate that the enhancement over the IGP hotspot can be attributed to anthropogenic emissions from the large cattle population and agricultural activities, especially rice production. Similarly, high $XCH_4$ values observed along the eastern coast can be attributed to the wetland emissions, as seen in Fig. 2f. Table S1 shows the mean observed $XCH_4$ and the variability over the entire study domain for the non-monsoon months of 2018 and 2019.

In general, the WRF-GHG simulations tend to show high bias in the winter months (see Fig. 8). A definite and widespread underestimation by the model was found in October 2018. However, in 2019, WRF-GHG was almost able to capture the observed $XCH_4$. In the summer months, the model shows patterns of overestimation in eastern India and underestimation in western India. These regional differences in patterns of $XCH_4$ can arise from heterogeneous sectoral distributions of surface emissions with seasonality that would have been misrepresented in the inventories in conjunction with the large-scale

meteorological influences (e.g. southwest monsoon over India, Chandra et al. (2017)).





While the peak total column $XCH_4$ for TROPOMI falls in October ($\sim 1870$ ppb), that of the WRF-GHG simulations is in November ($\sim 1883$ ppb). However, noteworthy is that both the model and observation indicate $XCH_4$ peak in either of the winter months between October and February. The WRF-GHG simulations show a higher variability (standard deviation) than observations for each month. WRF-GHG overestimates the $XCH_4$ values with a bias of $\sim75$ (72) ppb in 2018 (2019)(see Table

S1).

## 4.4 Assessment of the forward-model performance against surface measurements

As discussed in Section 2.3, we utilized hourly ground-based observations from a ground-based site, Thumba, to assess the WRF-GHG performance in the planetary boundary layer. The lowest level (approximately 35.2 m) of WRF-GHG-simulated $CH_4$ at Thumba is compared with surface-level observations of $CH_4$ and is presented here. Generally, the analysis indicated

a reasonable performance of WRF-GHG simulations. $CH_4$ mixing ratios are found to be lowest during the monsoon season (June–August), increasing from early October and peaking during the post-monsoon and winter months (November–January; see Fig. 9a). The maximum values are seen in December ($\sim 2100$ ppb). The hourly observations show high variability (about 112.7 ppb), but ranging from 1817.4 to 2612.6 ppb for 2018-2019; see Fig. S7. Despite some discrepancies, the WRF-GHG simulations broadly align with those highly fluctuating observation patterns, capturing about 56% of the observed variability.

Monthly averaged simulations and observations are found to be highly correlated in October despite their factor of magnitude differences (Figure not shown). The mean difference between observations and simulations is 47 ppb, but it shows large model-observation variability of up to 73.9 ppb (Fig. 9(a) and Fig. S7(a)). But, this large discrepancy between the model and observation can be caused by the influence of the fine-scale nocturnal coastal meteorological conditions prevailing in the measurement site as reported in Kavitha et al. (2018). Considering only the afternoon hours, the model-observation differences

are reduced to 6.4 ppb, with a maximum difference varying up to 28.1 ppb, capturing about 79% of observed variability (Fig. 9(b) and Fig. S7(b)). The above result is promising, confirming the usability of those afternoon measurements representing well-mixed atmospheric conditions, which can be utilized for future carbon assimilation systems in conjunction with a high-resolution forward modeling framework. As seen for all hours, WRF-GHG generally underestimates surface $CH_4$ mixing ratios (Fig. S7). Notably, the model-observation differences peaked in winter, owing to the unusually high variability seen

in the observations during this period. The effect of enhanced vertical mixing can be seen in the summer months, causing low observed $CH_4$ magnitude and associated mixing ratio variability. Noteworthy is that the magnitude of observed $CH_4$ is found to be the smallest during the summer monsoon. While the shallow planetary boundary layer (PBL) in the winter accumulates the effect of surface emissions to the lower boundary, increased boundary layer mixing in the summer can cause



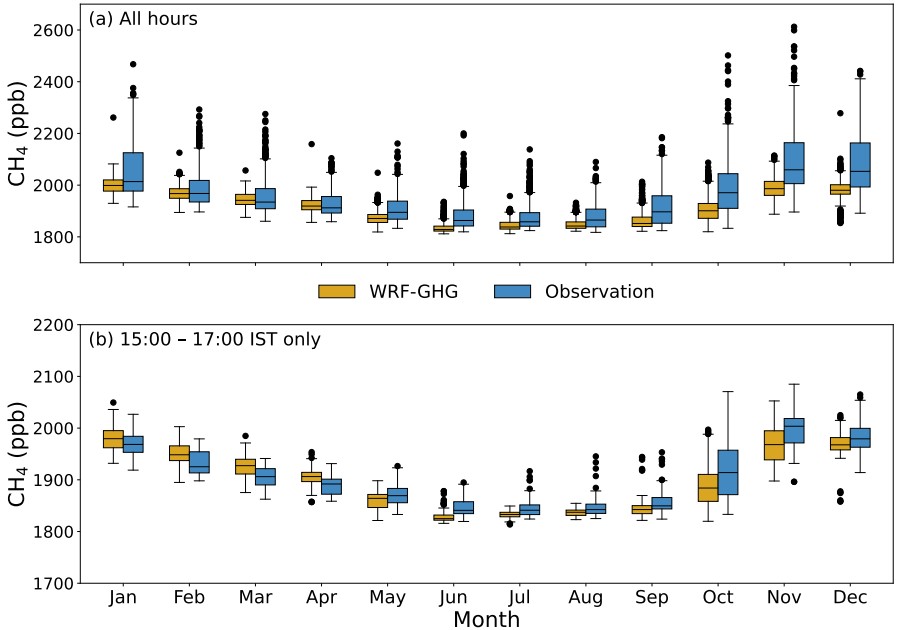

**Figure 9.** Monthly distribution of observed and WRF-GHG simulated $CH_4$ mixing ratios at Thumba for 2018 & 2019 for (a) all hours and (b) only 15:00 - 17:00 Local Time (IST hours) ($25^{th}$ and $75^{th}$ quartiles; see the site location as denoted in Fig. 1). Note that the ranges of the y-axes are not uniform in panels to improve visibility.

lower $CH_4$ magnitude and variability. Further, Guha et al. (2018) and Metya et al. (2021) report that the influx of clean air

from the Southern Hemisphere, carried by the monsoonal south-westerly winds, can influence the surface $CH_4$ to lower its

concentration. The high rates of OH radical oxidation may also influence surface $CH_4$ mixing ratios (Lin et al., 2015).

Even though WRF-GHG has shown reasonable performance while evaluating the ground-based observations from a complex site (located near the southernmost coastal boundary of the model domain), the robustness of the model needs to be

further examined with multiple locations across India when they are available. Those evaluations are particularly necessary for

assessing our confidence in the derived posterior fluxes.

### 4.5 National $CH_4$ budget estimation via inverse optimization

In this section, we present estimates of India's anthropogenic $CH_4$ budget for the period 2018-2019, derived through inverse

optimization as described in Sect. 3.2. The EDGAR emission inventory reports an annual mean $CH_4$ emission budget of 28.7

$Tg\ yr^{-1}$, and we assumed 80% uncertainty (23 $Tg\ yr^{-1}$) in our prior as discussed in section 3.1. The posterior annual emission

estimate is 24.3 Tg, with the uncertainty reduced to 3.3 Tg. The percentage of error reduction (calculated using Eq. 8) for

monthly posterior fluxes ranges from 68 to 92 %. Our inverse model results indicate an overestimation of 14 to 23 % in the



EDGAR inventory. Incorporating biomass burning emissions from GFAS has an impact of +0.3 Tg yr$^{-1}$ on both prior and posterior emission estimates over the Indian region. As per the India Fourth Biennial Update Report (BUR4) submitted to the United Nations Framework Convention on Climate Change (UNFCCC) (MoEFCC, 2024), the CH$_4$ emission budget for

India is approximately 19.6 Tg yr$^{-1}$, which is around 32 % less than the EDGAR reported emissions during the 2018-2019 period. Previous studies also reported an overestimation of the global emission inventories over India. The Global Methane Budget 2000-2020 (Saunois et al., 2024) reported 21.7 Tg of anthropogenic CH$_4$ emission from India. The total CH$_4$ emissions derived from a combination of satellite data (GOSAT), surface and aircraft measurements, and the atmospheric transport model for 2010–2015 were found to be 22 Tg yr$^{-1}$, which is substantially lower than the emissions reported by the EDGAR v4.2

inventory (Ganesan et al., 2017). Janardanan et al. (2024) reported the annual averaged (2009 - 2020) CH$_4$ emissions from anthropogenic sectors over the India as 24.2 ± 2.1 Tg yr$^{-1}$ which is close to our results. On the other hand, Raju et al. (2022) reported that the CH$_4$ budget for peninsular India is 0.13 Tg yr$^{-1}$ higher than inventory-based estimates for the period 2017-2018. These variations in emission reports emphasize the need to improve CH$_4$ emissions in India.

      Although we utilize high-density, high-quality, and high-resolution TROPOMI satellite retrievals together with a high-
resolution transport model, our first-order inversion algorithm is limited by its dependence on the spatial distribution of emissions in prior inventories. Prior emission errors in the spatial distribution, which would have propagated from data sources or bottom-up methodology, cannot be minimized in our optimization procedures. However, we expect that those spatial errors may have a minor impact on our annual national estimates owing to our temporal and spatial averaging. Though the GFAS inventory includes agricultural residue burning, small fires that are common in smallholders for clearing the wastes and field

preparation can be missed from prior inventories, as reported in Deshpande et al. (2022).

      We excluded natural wetland emissions from the inverse optimisation as we focused on major anthropogenic emission sources. Further, the natural wetland prior emissions have resulted in negligible impacts on the column mixing ratio enhancement (figure not shown), which is smaller than the uncertainty of the satellite measurement. However, Indian wetland emissions also vary based on inventories and prior models. Approximately 7.5 Tg CH$_4$ emissions from the Indian region were reported in

the Global Methane Budget 2000-2020 (Saunois et al., 2024). Janardanan et al. (2024) used wetland prior emissions from the Global Methane Budget 2000-2017 (Saunois et al., 2019) in their inversions and reported approximately 3.8 ± 0.16 Tg CH$_4$ emissions annually from Indian wetlands. At the same time, the BUR4 report (MoEFCC, 2024) has not included the wetland emission estimates, possibly due to inadequate data coverage. Bernard et al. (2025) also discussed the limitations in modeling the wetland emissions from the tropical region due to the inadequacy of available measurements. The above level of estima-

tion discrepancies calls for a country-specific wetland inventory that can also be used as reliable prior fluxes in future inverse





modeling. Another limitation can be the possible overlapping of natural and anthropogenic (agricultural fields) wetlands in the emission inventories used, which may overestimate the sectoral contribution of posterior fluxes.

## 5    Conclusions

In this study, we investigate the potential of TROPOMI satellite observations along with a high-resolution atmospheric transport

model, WRF-GHG, to represent the distribution of $CH_4$ emissions over the Indian region. Analysis of the bottom-up inventories shows enteric fermentation as the most significant contributor to $CH_4$ emissions in India (42.9 %), followed by wastewater treatment (19.2 %), agricultural soil (12.4 %), fuel exploitation (6.7%), and wetlands (5.2 %, excluding agriculture). The above proportions highlight the considerable impact of anthropogenic sources on $CH_4$ accumulation in the atmosphere. As expected, $CH_4$ emissions from rice agriculture (August), wetlands (July), and biomass burning (March) exhibit distinct seasonal patterns.

The bottom-up anthropogenic $CH_4$ emissions, and consequently the total atmospheric $XCH_4$ mixing ratios, have shown some peaks over South India due to a few prominent emission hotspots.

The WRF-GHG simulations of $XCH_4$ mixing ratio enhancements indicate considerable contributions from anthropogenic and biomass burning emissions, particularly in the IGP region (>60 ppb). The highest seasonal enhancements of anthropogenic $XCH_4$ occur during winter, influenced by agricultural emissions, biomass burning, and atmospheric winter transport. Both the

observed and modeled total $XCH_4$ show significant peaks over the IGP region, with values ranging from ~1862 to ~1870 ppb during October-November. Though WRF-GHG remarkably captures atmospheric $XCH_4$ patterns, simulations generally overestimate $XCH_4$ levels compared to TROPOMI. The total $XCH_4$ along the eastern coast suggests the effects of wetlands on atmospheric column mixing ratios. Our high-resolution model is capable of capturing surface $CH_4$ variability, especially for the well-mixed conditions, as confirmed by the ground-based $CH_4$ observations. Such ground-based observations across India

are essential for evaluating the full potential of high-resolution models in representing the atmospheric distribution of trace gases.

The inversion analysis using our high-resolution model and TROPOMI observations reports an annual mean anthropogenic $CH_4$ emission budget of $24.3 \pm 3.3$ Tg yr$^{-1}$ (excluding biomass burning of 0.3 Tg yr$^{-1}$). Our estimations are 14 to 23 % lower than the EDGAR emission estimates. At the same time, our estimate is 19 % higher than what the Government of India reported

to the UNFCCC for the same period but close to the latest Global Methane Budget 2000-2020 (21.7 Tg). We emphasize the need for a robust reporting of $CH_4$ emissions from the Indian region in the global emission inventories, which also require more ground-based atmospheric trace gas measurements along with the advancement of satellite capabilities and atmospheric tracer

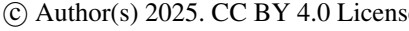

transport models. Overall, the analyses highlight that TROPOMI observations can offer valuable insights into $CH_4$ emissions, and the WRF-GHG model has the potential to be used in the assimilation system to refine the emissions.

*Code and data availability.* The anthropogenic $CH_4$ emission inventories used in this study are downloaded from https://edgar.jrc.ec.europa. eu/archived_datasets (last access: March 2024) (Crippa et al., 2024). CAMS global biomass burning emission based on fire radiative power (GFAS) is accessed from Copernicus Atmosphere Monitoring Service (CAMS) Atmosphere Data Store, DOI: 10.24381/a05253c7 (last access : November 2023). The global wetland $CH_4$ emissions, WetCHARTs v1.3.1 is prescribed from https://daac.ornl.gov/cgi-bin/ (last access: November 2023) (Bloom et al., 2017). The WRF source code is freely available and can be accessed from https://www2.mmm.
ucar.edu/wrf/users/download/get_source.html. The TROPOMI/WFMD v1.8 product is made available via https://www.iup.uni-bremen.de/ carbon_ghg/products/tropomi_wfmd/.

*Author contributions.* DP designed the study, TAM and DP performed the model simulations, raw data analysis, and postprocessing, and wrote the initial version of the manuscript. JS, MVD, VT, and AR contributed to the data curation and figures. MB and OS contributed to data archival and processing. SBK contributed to editing. SS, IAG, and SB contributed to the ground-based data collection and pre-processing.
All authors contributed to the data analysis, interpretation and writing.

*Competing interests.* The corresponding author has declared that none of the authors has any competing interests.

ther geographical representation in this paper. While Copernicus Publications makes every effort to include appropriate place names, the final responsibility lies with the authors.

*Acknowledgements.* This study has been supported by funding from the Indian Ministry of Education and the Max Planck Society in Germany, which has been allocated to IISERB. The University of Bremen team acknowledges funding from ESA via project GHG-CCI+ (ESA contract no. 4000126450/19/I-NB) and the Bundesministerium für Bildung und Forschung within its project ITMS (grant no. 01 LK2103A). We acknowledge the support of IISERB's high-performance cluster system for computations, data analysis, and visualization.



The TROPOMI/WFMD retrievals were performed on HPC facilities funded by the Deutsche Forschungsgemeinschaft (grant nos. INST 144/379-1 FUGG and INST 144/493-1 FUGG). This publication contains modified Copernicus Sentinel data (2018–2019). Sentinel-5 Precursor is an ESA mission implemented on behalf of the European Commission. The TROPOMI payload is a joint development by ESA and the Netherlands Space Office (NSO). The Sentinel-5 Precursor ground segment development has been funded by ESA and with national contributions from the Netherlands, Germany, and Belgium. Thara Anna Mathew acknowledges the financial support provided by the Prime Minister's Research Fellows (PMRF) Scheme for providing a fellowship for a PhD. Jithin Sukumaran acknowledges the Council of Scientific and Industrial Research (CSIR) funding for his PhD fellowship. Imran A Girach acknowledges Prabha R Nair, former scientist at SPL, for supporting the surface trace gas measurements at Thumba utilized in this study.



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
