# Peer review of "Leveraging TROPOMI observations and WRF-GHG modeling to improve methane emission assessments in India"

_EGUsphere, 2025_

## Referee Comment (RC2)

**Review of Mathew et al. 2025**

https://egusphere.copernicus.org/preprints/2025/egusphere-2025-1977/

**General comments**

**Summary of the study**

This study is about estimating methane emissions from India by optimizing bottom-up emission estimates using TROPOMI data in an inverse modeling framework.

In section 1 (introduction), the significance of the research question, the state of research, knowledge gaps and goals of the study are explained.

Sections 2 and 3 introduce data and methods: the datasets analyzed in this study, the atmospheric transport model, and the prior emission datasets, the sampling method and the inverse modelling framework for emission estimation.

Section 4 (results and discussion) is organized as follows. Section 4.1 describes the spatiotemporal distribution of emissions in the bottom-up datasets introduced in section 2. Section 4.2 describes the spatiotemporal distribution of XCH4 modeled using the bottom-up datasets and a boundary condition dataset. Section 4.3 and 4.4 add comparisons of the modeled mixing ratios to observed XCH4 from TROPOMI and surface CH4 from one ground-based site (Thumba), respectively. Section 4.5 shows the inverse modeling results.

The study concludes with section 5, which states the conclusions drawn from the analyses.

**Overall approach**

Overall, the study is a contribution towards an important goal, namely improving the understanding of atmospheric methane emissions from India. The methodological approach (inverse modeling of atmospheric methane transport to optimize a prior flux estimate) is well established in the literature. That and the comparisons to similar results from the literature provide confidence that the results are sound.

**Selection of shown results**

The study puts a big emphasis on describing the spatiotemporal distribution of methane emissions from published bottom-up datasets (section 4.1, 4 pages), as well as their impact on XCH4 (section 4.2, 2 pages). In my opinion, the study (abstract/introduction/conclusion) should explicitly state that such a review is one goal of the paper. Because for the goal of the study as stated in the title, "to improve methane emission assessments" using an inversion, the level of detail of these sections is too much, in my opinion. At the same time, the statements in section 4.2 and 4.3 about the separate contributions of the different sources (anthropogenic, biomass burning, wetlands) to XCH4 would actually benefit from a bit more detail, namely showing separate plots of these contributions and of the background (details in the specific comments).

By contrast, the section on emission estimation results - which should be the major part of a study that aims "to improve methane emission assessments" - is very short (section 4.5 - 1.5 pages): The only result given is the annual national total emissions and their uncertainty, and a comparison to the corresponding bottom-up estimate. Many questions one would expect to be addressed - and that could be addressed - using the results of the already performed inversion are not addressed. I recommend that the authors show more details of their results in section 4.5, i.e. the spatiotemporal and sectoral distribution corresponding to their state vector resolution, and whether/how much the fit to the observations (TROPOMI and Thumba) improves after the optimization. If there is a good reason why the authors wish to not show such additional details, it needs to be stated in the manuscript, and the expectations of the reader should be managed, for example by adjusting the title (e.g. by switching to a wording with "_towards_ improving methane emission estimates").

**Language**

A general language improvement is required:

- The word "the" is overused. Some, but not all occurrences, are given in the specific comments below.

- In many cases, the wording is unclear to me or appears to be inaccurate. In many cases where I found the wording confusing, it may be due to the usage of comparatives ("more", "growing", "enhanced", "increased") when positives ("large", "high") seem to be the appropriate choice to me. Unclear wording is what a lot of the specific comments below are about, and in many cases, I added what I think the statement should rather be.

**Specific comments**

**Abstract**

- Line 5: Please adopt the correct definition of "WRF" throughout the document ("Weather Research and Forecasting" model)

- Line 7: "non-negligible uncertainties" - should rather be something like "non-negligible differences", no?

- Line 8-9: "We find that the WRF-GHG simulations overestimate the XCH4 and underestimate the near-surface CH4 distributions".

     - This sentence overstates the representativeness of the single ground-based measurement site used in the analysis. It is even a point later in the conclusions that CH4 emission estimation in India would probably benefit from expanding the observation network. The authors should harmonize abstract and conclusions on this point.

     - Also, since this observation is so prominently placed in the abstract, I expected an explanation or at least a discussion of this observation. For example, it could hint at errors in the vertical transport, but also simply reflect the different representativeness of the column- and ground-based observations used here. Neither was investigated. Instead, the conclusion section only states (line 423-424): "Our high-resolution model is capable of capturing surface CH4 variability, especially for the well-mixed conditions, as confirmed by the ground-based

CH4 observations". Thus, the significance of these differences stated here in the abstract remains unclear, and abstract and conclusions need to give the same message on that point.

- Line 9: I'm not familiar with the term "first-order inversion". So I think it should be explained in the text, replaced with a clearer term or removed (the method used is an "inversion" - I see no need for an extra qualifier like "first-order"). After reading the manuscript, I think "first-order" refers to the fact that the state vector is rather coarse.

- Line 10-11: "showing that the current global emission inventories overestimate CH4 emissions considerably" - please state which inventories and by how much they overestimate the emissions.

**Introduction**

- Line 19: Please cite a more recent reference about recent atmospheric greenhouse gas burdens

- Line 21: If possible, please cite a more recent reference about the lifetime of atmospheric methane

- Line 28-29: This statement on OH trends is not accurate. First of all, contrary to what is stated here, the cited reference Stevenson et al. 2020 concludes that OH _increased_ recently (from 1980 to 2014). But also, as far as I'm aware, there is no definite consensus on the OH trend. E.g. Thompson et al. 2024 see no trend and cite studies with opposing results (https://acp.copernicus.org/articles/24/1415/2024/). There is also more context in Saunois et al. 2025 (https://doi.org/10.5194/essd-17-1873-2025) and Zhou et al. 2024 (https://doi.org/10.1088/1748-9326/ad4b47).

- Line 50-51: Please clarify in the text what you mean by "inadequate modeling systems"

- Line 52: "imminent" is not the appropriate term here

- Line 70-71: "We demonstrate the advantage of ... operating at a high resolution comparable to TROPOMI, which may minimize forward model-related uncertainties in the carbon assimilation system over India". This sentence should be rephrased: it suggests that the authors experimented with varying transport model resolutions, which is not shown in the manuscript. The model also doesn't have that much higher resolution than others cited here (e.g. Ganesan et al. 2017 ran at 12-16 km), and the study does not address whether the forward or inverse model results shown here are superior to those of previous studies.

- " Section 4 discusses the results of the study from the data analysis conducted" – I'd remove "from the data analysis conducted"

- Line 71: WRF definition (see comment above)

**Section 2**

- Line 106: Please add the citation for ERA5 here (or cross-reference to Table 1)

- Line 110: Provide a citation, link and version of the CAMS product used here (or cross-reference to Table 1)

- What's the model top of the WRF simulation (according to Thilakan et al. it's 50 hPa)? Please add this to Table 1. How do you extrapolate the model profile to the top of the

atmosphere to compare with TROPOMI? I.e., does your comparison method account for the CH4 decrease in the stratosphere?

- Line 112: "far-field (background)": I find the term "far-field" unclear, as it can mean different things depending on context. I suggest to remove the term "far-field" and only refer to "background" or "initial and boundary conditions". Here and in Section 4.2. Similarly, the term "regional sources", used elsewhere in the manuscript, should be properly defined (along the lines of "sources within the simulated domain")

- Line 123: Kretschmer time profiles: mention what does this entail - e.g. month + week day + time of day?

- Section 2.3 - Thumba validation station: Add something about the area it represents or at least acknowledge that it's not representative of the whole of India. "To assess the model's performance at the surface level" is overstating the representativeness. I found something on that in Uma et al. 2024, you could briefly summarize those findings here.

**Section 3**

Section 3 only has one subsection, section 3.1. In my opinion, the content of section 3 belongs to the methods, i.e. I would integrate it into section 2.

- Line 162ff: "The quantities to be optimized, represented by the state vector x with n elements $x_1, x_2, x_3, ..., x_n$ correspond to the total monthly emission information averaged over each political state in India." The term "total monthly emission information" appears to be inaccurate, because (1) wetland emissions were not optimized (line 182ff) and (2) the EDGAR and GFAS fluxes were optimized separately. Or not? Please adjust the text so it becomes clear what exactly was optimized.

- Line 170: F(x) is not only WRF-GHG but also what is sometimes called "flux model" (in your terminology: mapping of x to phi) and here, the mapping of WRF-GHG's output to total columns (Eq. (1)). Please make this clear in the text. Suggestion: Write out the relationship between phi (emission field) and x (state vector element) as an equation.

- Line 177: "the perturbed emissions"- remove "the", I thought I had missed the introduction of the perturbation. Also, specify how you perturbed the fluxes. From the equations I infer: You constructed response functions for each state vector element by adding a small flux to the prior of the state vector element. Or was it a multiplication by a 1 + a small number?

- Eq. (4) - Define "TR"

- Eq. (5) Error in the last term ("x_A" should be "x")

- Line 182ff: Earlier, the authors write that the prior represented anthropogenic, wetland and biomass burning emissions. Here, it sounds like only anthropogenic and biomass (add "burning") emissions were optimized, not wetlands. I remarked on line 162 that this it needs to be clarified what exactly was optimized separately. In addition, the authors need to state why they can omit the optimization of wetland emissions, i.e. neglect their uncertainty that was cited in the introduction.

- Line 185: I would suggest to choose one term for fluxes inside the model domain and define it properly. Both "regional" and "local" is used in different places in the manuscript, and I wondered if different things are meant.

- Eq. (9): summation symbols missing

- Line 213: Here is stated that Edgar v8 is used, earlier it was v7 - which is it? Correct all occurrences.

- Line 214: Remove "the" from "the emission activities"

- Line 220: Remove "the" from "the wetland" and "the agriculture"

- Line 220-221: Please clarify the sentence "The seasonal sources of CH4 include agriculture (see Fig.2 (b)) and biomass burning". I presume you mean something like "The sources with significant seasonality include agriculture and biomass burning"?

- Line 222ff: "The positive anthropogenic trend in CH4 emissions in India can be expected owing to the large cattle population and agricultural activities (especially rice production and waste management) as indicated by 2010-2015 GOSAT observations (Maasakkers et al., 2019)". This sentence is confusing. Do you mean to say something like "A large anthropogenic contribution to the total CH4 emissions in India can be expected …" ? The word "trend" usually refers to multiannual trends, but is used here to describe a figure that shows one year of results. Also, a "positive trend" is not explained by "large" agricultural activities as stated here, only if they were "growing". Please clarify.

- Line 229: Fix this reference for Edgar 4.2; in the text it's cited as "Commission et al.", and in Line 514 it shows up as "Commission, E. E. et al."

Line 232: State which months are the "summer monsoon season". EDIT: Later in the manuscript, I also got confused which months are defined as winter. Please add season labels to Figures 3 and 5.

Figure 1: It's hard to see the delineation of the regions (IGP, ...) because at the boundaries, only one of the lines can be seen. Please modify the figure so that the colored borders are visible for all boundaries. Maybe dashed lines work.

Figure 2:

- Add the region boundaries as in Fig. 1.

- Why not include a panel for the coal mining sector? The authors even discuss a hot spot from coal mining (Sect. 4.1), but it's not in any figure

**Results and discussion**

**Section 4.1**

- Line 233: "the least" -> "the minimum"

- Line 233: I find the usage of percentages from here on out in this section confusing: are they the share of monthly total fluxes? Or perhaps shares of the flux in a specific sector of the annual flux of that sector? The text seems to be about seasonality of the fluxes, but if the stated numbers are percentages of the monthly total fluxes, that seasonality is not necessarily the same as the seasonality of the fluxes. Please clarify.

- Line 247 and 248: Define the regions (here: "SI" and "NI") upon first use in the text (I know they are defined in a figure caption, but as a reader, I don't want to go searching).

- Line 253: Please state that these are cities.

- Lines 252ff: I can't find these hotspots on the maps. Can you mark them?

- Line 256: "has the issue of ... possibly generating more volatile gases": First, reformulate "has the issue of" in a neutral way. Second, "More volatile gases" compared to what? Do you mean "Large amounts of volatile gases"? Third, if correct please change from "volatile gases" to "methane".

- Line 257: "increased agricultural rice emissions" - increased compared to what? I guess the authors mean "high" emissions?

- Line 271: Define "NEI" (first usage in the text)

- Line 272: an order "of magnitude"?

- Lines 278-286: These considerations are the motivation for doing the present study. In my opinion, they belong in the introduction.

- Lines 282f: "Though inverse modeling can offer better scope in updating the CH4 budget, those approaches are limited by the insufficient coverage of mixing ratio observations and inadequate representation of the process and spatiotemporal distributions in the forward models."

    - I think this sentence is missing a causal connection. And it's unclear to me what "can offer better scope" means. My suggestion would be: "Though inverse modeling can improve the CH4 budget, its ability to overcome biases in the bottom-up models used as prior estimates can be limited by insufficient coverage of mixing ratio observations".

**Section 4.2**

- Line 289ff: Throughout this section, it's not clear to me which sources inside the model domain are considered for this XCH4 analysis: The title suggests it's only the "anthropogenic" sources, i.e. excluding biomass burning and wetlands, according to the terminology established in the methods section. However, this sentence here seems to imply that also the other sectors (biomass burning and wetlands) are included. In Fig. 4, it's yet something else: anthropogenic + biomass burning. Also, sometimes "anthropogenic + biomass burning" seems to be meant when "anthropogenic" is written (I may be wrong about that one). I get that biomass burning is at least partly from human activity in the study region, as explained a few sentences below. The text reads like biomass burning is mostly or completely anthropogenic, but that needs to be stated. Please clarify this section, e.g. by sticking to the terminology of the different sources as established in the methods section.

- Lines 289ff again: Also, if the XCH4 from wetlands is indeed not shown in this section: it's stated in lines 327-329 that the wetlands do make a contribution to the XCH4. So this needs to be shown in a figure.

- Line 290: "biomass" -> "biomass burning"

- Line 291-293: The sentence "The IGP and surrounding regions exhibit significant XCH4 enhancements (>60 ppb) from background contribution (far-field fluxes) attributed to anthropogenic and biomass-burning fluxes (see Fig. 4)" has multiple issues:

- In the previous sentence, the authors write that they discuss XCH4 enhancements due to regional sources here. Therefore, I think they meant to say "regional sources" instead of "background contribution (far-field fluxes)" here as well, otherwise the sentence doesn't seem to make sense.

- What is the significance of the number ">60ppb"? Nowhere in Fig. 4 or 5, which show the enhancements described here, does this look like a special value - it's the upper limit of the colorbar of Fig. 4, but seems to be an arbitrary cut-off. Why not state the exact numbers like "up to ..." or "from ... to ..."?

- The sentence "Seasonally, compared to summer, the highest XCH4 enhancements occur during winter (with a maximum of ~ 251 ppb in January)" and Fig. 4 or 5:

  - The maximum of 251 ppb is printed as "max" in Fig. 4 for January, and can presumably be seen as an outlier in Fig. 5. So I assume the two figures show the same "max" (it's not stated what the "max" is - is it snapshots as reported by WRF? Hourly averages? Something else?). However, in Fig. 5, there are even higher outliers in Oct-Nov, whereas the "max" in Fig. 4 among these two months is shown as 142.2 ppb. So it seems that Fig. 4 and 5 are inconsistent.

  - Line 293: "Seasonally, compared to summer" sounds confusing. I guess "compared to summer" can be deleted.

  - State to which region the statement refers to - I guess all of India?

- The sentence "The minimum enhancement is seen in the monsoon time, possibly due to large-scale mixing in the atmosphere" has multiple issues

  - What does "large-scale mixing" mean? It's not a term I'm familiar with and it's not explained in the manuscript. I guess higher wind speeds? Or do you mean higher boundary layers? Vertical transport alone doesn't affect total column measurements much, so this can't be the full story. Perhaps higher boundary layers in combination with higher wind speeds in upper layers. Please clarify. Also, please remind the reader of the seasonality of the fluxes here.

  - Specify for which region this statement is true! In Fig. 5 I can see that it's not true for South India, which is affected by monsoon. So I don't understand the given explanation.

- Line 300: The phrase "high spatial distribution" doesn't make sense – please adjust.

- Fig. 4:

  - This figure doesn't convey much useful information: all the panels look the same, and only the numbers in the panel titles are referred to in the text (and they also show up in Fig. 5). Why not show an annual mean map here, and move the monthly panels to the supporting information.

- Fig. 5:

  - No outliers are plotted for the South India region. Is that correct?

**Section 4.3**

- Line 308: Do the authors mean "reasonable _amounts_"?

- Line 310f: Please clarify "enhanced". E.g., provide context for the term "enhanced": "enhanced values ... compared to ...". Or change to a neutral term like "highest mixing ratios".

- Line 311-313: As explained above, I think a shallower vertical distribution of emitted tracers in shallow PBLs alone should not cause large gradients in total columns from TROPOMI, as its sensitivity is rather (though not completely) uniform across the lower atmospheric levels.

- Line 313: The use of the term "increased" surface emissions is unclear to me: October-November is not the seasonal emission maximum, according to Fig. 2b.

- Line 313f: "These increased local emissions, also seen in Fig. 3, are typical for some parts of India during the October-November season". State explicitly which parts of India and which flux components are meant here. The reference to Fig. 3 is not enough, because it's too much information to process. And in section 4.1, where seasonality is discussed, I don't see a statement about an October-November maximum either. I guess this statement might refer to biomass burning emissions in the IGP region, as mentioned two sentences later in context with aerosol load (still, state what is meant here) - however, their contribution to total emissions is tiny compared to the anthropogenic emissions, according to Fig. 3. So from the data presented, it's unclear to me whether there actually is an emission peak somewhere in this season – so it remains unclear to me what causes the large XCH4 values in Fig. 6 that is referred to here.

- Line 314: "... the likelihood of increased aerosol loads, ..." -> "the likelihood of bias in the observations due to increased aerosol loads, ..."

- Line 322f: "We find that the WRF simulations generally overestimate the total XCH4 mixing ratio over the Indian region compared to TROPOMI observations, peaking in winter months (>1850 ppb) (see Fig. 7)."

    - The choice of highlighting the number ">1850 ppb" is unclear to me. Before, it was stated that the mean observed concentration went up to 1870 in that period, Fig. 7g states 1883.2 ppb, so why cite 1850 ppb for overestimation by the model.

    - Clarify: do you mean that the WRF results peak there or the difference between model and observations?

- Line 324: I stumbled upon the phrase "suggesting the increased impact". Perhaps "suggesting a large impact" is meant. However, statements on contributions to XCH4 by sector (anthropogenic, biomass burning, wetlands) could be made much better when showing plots of all three (as suggested in my next comment).

- Line 327-328: "Similarly, high XCH4 values observed along the eastern coast can be attributed to the wetland emissions, as seen in Fig. 2f." I have several comments about that statement:

    - In my opinion, the eastern coast doesn't stick out with particularly high observed XCH4 in Fig. 6. Perhaps in October and November, but not really in April and May. Clarify to which season/months the statement applies.

    - Fig. 2 shows that "agricultural soils" have, on average, larger emissions than wetlands along the eastern coast. Therefore, I expect higher contributions of this sector than from wetlands to XCH4 in that area. So the conclusion by the authors that XCH4 peaks there are explained by wetland emissions is, in my opinion, not convincing.

    - If the wetland emissions had a big influence on parts of the observational dataset, it would be unclear why they were excluded from the optimization. In lines 396-398, it is even

stated that the influence of wetland emissions was very small overall - contradicting the statement here.

       - There are a few options how the authors could address this comment. They could clarify which emission dataset has the higher influence on XCH4 on the eastern coast by running them as separate tracers in the WRF model, and show wetland XCH4 contributions analogous to Fig. 4. The abovementioned lines 396-398 suggest that the authors already did run separate tracers, so adding these plots to the manuscript or supplement would not be a lot of work while adding a lot of value to this section. Or they could clarify which months they mean, and state that the "agricultural soil" dataset could fit the observed XCH4 distribution too - or explain why that is not the case.

- Line 338: I feel like it's a bit of a stretch to classify October as "winter", and October - February are not classified as one season in Fig. 3. As I suggest in another comment, please add labels for the seasons to the plots.

Figure 6 vs Fig. 7:

- It looks as if the data coverage is different for the observations than the simulation. E.g. in February over the Tibetan plateau, it looks like there are much fewer data points in Fig. 6 than in Fig. 7. Was the data selection of model and observations identical for their comparison? Is it just that data points below the lower bound of the colorbar are shown in white? In that case, choose a different color than white for the lower bound to distinguish from missing data.

**Section 4.4**

- Line 350: This sentence is unclear: "Monthly averaged simulations and observations are found to be highly correlated in October despite their factor of magnitude differences (Figure not shown)."

- Line 355: Here, the performance of the model is evaluated against observations in terms of total mixing ratios. The fact that the model captured "79% of observed variability" is dominated by the seasonal cycle of the boundary conditions, so is not related to the accuracy of fluxes in the domain. It could be more informative to do this comparison after subtracting modeled boundary conditions from both datasets (although I acknowledge that the boundary conditions may have biases too).

- Line 360-366: Here the authors cite three possible mechanisms for the low variability and means of the mixing ratios during summer: (a) enhanced vertical mixing, (b) influx of clean air and (c) OH oxidation. The vast majority of the joint effect of (b) and (c) together could be gleaned from the contribution of a background tracer (i.e. CAMS). Hence, the suggestion in my previous comment, i.e. to separate the influence of the background, could provide insight here as well.

**Section 4.5**

- Line 377f: "Incorporating biomass burning emissions from GFAS has an impact of +0.3 Tg yr-1 on both prior and posterior emission estimates over the Indian region". Please clarify: Does this mean that there were two simulations, one in which the GFAS emissions were held constant and one where they were optimized?

- Line 387: Which inventory? Also EDGAR? What were the estimated emissions?

- Line 388: "improve CH4 emissions in India" -> "improve CH4 emission estimation in India"

- Line 391-392. "Prior emission errors in the spatial distribution, which would have propagated from data sources or bottom-up methodology, cannot be minimized in our optimization procedures". I disagree. They were optimized, just with coarse resolution.

- Line 393-395: This closing sentence has nothing to do with the rest of the paragraph - please reorganize.

- Lines 396ff: In my opinion, the statements on the various emission estimates in the literature belong in the introduction.

- Line 396-398: "We excluded natural wetland emissions from the inverse optimization as we focused on major anthropogenic emission sources. Further, the natural wetland prior emissions have resulted in negligible impacts on the column mixing ratio enhancement (figure not shown), which is smaller than the uncertainty of the satellite measurement." I would remove the first sentence - it's not a justification why excluding wetlands from the optimization would be reasonable in this case. Also, as I suggested above, it would add a lot of value to show this figure. Finally, the statement contradicts the above discussion of how wetlands allegedly have a noticeable influence on observed XCH4 at the eastern coast (see my comment on that above).

- Line 399: "However, Indian wetland emissions also vary based on inventories and prior models" - unclear. Do you mean they "vary among bottom-up models"?

- Line 399: "Approximately 7.5 Tg CH4 emissions from the Indian region ..." -> "Approximately 7.5 Tg CH4 _wetland_ emissions from the Indian region ..."

**Section 5 (Conclusions)**

- Line 418-419: "The highest seasonal enhancements of anthropogenic XCH4 occur during winter, influenced by agricultural emissions, biomass burning, and atmospheric winter transport". As stated above, I think the authors did not sufficiently demonstrate how exactly winter transport (i.e., shallow PBL height) influenced the total columns of CH4. I think they could do so by calculating mean wind speeds in the atmospheric boundary layer, which are the winds that the emitted methane is transported with, and may find that they are higher in summer presumably because the boundary layer reaches higher. Or cite a study where this was shown.

- Line 422-423: "The total XCH4 along the eastern coast suggests the effects of wetlands on atmospheric column mixing ratios." As stated above, I am not convinced that it isn't rather the agricultural soils. As also remarked above, this statement contradicts the statement elsewhere in the manuscript that wetland emissions had a "negligible" effect on XCH4.

**References**

- Many links follow the pattern "https://doi.org/https://doi.org/<doi>". While they seem to work, this is probably not intended.

---

## Author Comment (AC1)

We appreciate the editor and two anonymous referees for their time and constructive review of our manuscript. We greatly value the feedback received, have thoroughly reviewed and addressed all comments, suggestions, and concerns, and implemented necessary revisions into the manuscript. We believe that these modifications have significantly enhanced the clarity, accuracy, and overall readability of our manuscript - Thank you.

The modified texts in the revised manuscript are given in *blue italic* font.

**RC:** Referees' Comment (in black font)

**AR:** Authors' Responses (in blue regular font)

**Detailed Response to Referee #1**

**General Comments:**

**RC:** Choice of inversion method: Why was an analytical inversion chosen instead of an ensemble Kalman filter, which would allow for a much finer resolution of the state vector? India's political states are quite large, and it would be valuable to assess the redistribution of emissions relative to the prior also at a sub-state scale.

**AR:** Thank you for the comment. We acknowledge that different inverse techniques exist depending on observational type/density and spatiotemporal scales of state vectors. We agree that ensemble-based approaches like the Ensemble Kalman Filter (EnKF) offer potential for improving state vectors at finer scales, especially advantageous while utilizing large sets of observations (without the demand of an adjoint model). Our decision to use an analytical inversion was driven by the following factors: First: EnKF is often opted for a dense observational network and is a better choice in order to solve large inversion problems. While refining sub-state level state vectors would be beneficial with the extent of satellite-based column observations of methane, the region is limited with the availability of ground-based measurements that contain more information on near-field flux influences. We believe that the information derived from the assimilated observations, as done in this study, may not be adequate for a reliable reporting of emissions at finer scales; hence focusing primarily on deriving emissions on a regional scale in the present study. Second: An analytical solution is feasible for linear tracer flux mapping, like for methane, providing error statistics and information (note that EnKF approximates error statistics based on ensembles), especially for the dimension of the inversion problem (again limited by the observation space) analyzed in the present study. The analytical inversion, as applied here, ensures computational efficiency much better

than EnKF and interpretability while incorporating state-wise spatial constraints, effectively capturing nationwide emission redistribution.

At the same time, we acknowledge the advantage of EnKF and the value of sub-state space resolution in the inverse-based studies. By establishing more comprehensive ground-based observation networks that offer accurate and precise measurements, in addition to column observations and high-quality atmospheric transport models, future investigations can explore various inverse methodologies such as the ensemble Kalman filter (EnKF) and 4D variational inversion (4D-Var), which are capable of managing highly resolved state vectors, resulting in improved emissions at a significantly finer sub-scale.

To explicitly represent this point, we revise the conclusion as follows- Line 492-495:

"With the advancements of denser observational coverage and the high-quality atmospheric transport models, future research can thus explore and evaluate different inverse techniques like ensemble Kalman filter (EnKF) and 4D variational inversion (4D-Var) that can handle highly resolved state vectors, leading to improved emissions at a much finer sub-scale."

**RC:** Use of observations: It is unclear how exactly the observations are used in the inversion. On page 9, line 164, the authors write that the measurement vector y contains m elements, representing the total column observations where m represents the total number of observations in each political state. Does this imply that only observations at a given state are used to constrain emissions within that same state? Ideally, all observations should be used in the inversion. If this is only a wording issue, the sentence should be rephrased to avoid confusion.

AR: All available observations across the domain were used simultaneously in the inversion process to constrain emissions. The phrase "m observations in each political state" (now Line 180) was intended to describe the spatial grouping of observations within the target region, which is used for constructing Jacobian matrices. With this set-up, the inversion primarily captures near-field observational influence by giving negligible weights to far-field signals. Also, it is expected to minimize spatial inconsistency in the inferred emissions while avoiding over-interpretation of distant observational signals. Please note that the scope of this study is to interpret the influence of regional fluxes (>100 km).

**RC:** Presentation of results: The inversion results are currently presented only as total national emissions. This is insufficient. The changes in emissions at the state level should be shown. A map illustrating the posterior emissions by state would be very helpful.

AR: Thank you for the comment. Although sub-national scale data would be ideal, the focus of the present study is to provide national-scale anthropogenic CH4 emission estimates, considering the coverage of the satellite observations, with the limited ground-based measurements (including column observations) in India. The inversion framework we employed in principle allows for posterior emission analysis at finer spatial and sectoral scales. However, we chose to report only the aggregated national totals, as we expect that the information content deduced from the available assimilated observations can be insufficient to obtain the robust reporting of the sectoral or sub-scale emissions. With denser observation networks, we aim to refine the inversion setup and report state-wise emissions in the future. For instance, Pillai et al. (2016) highlight the limitations of satellite-based inverse modeling to derive highly resolved emissions due to insufficient observational density, suggesting that denser observation networks could enable more accurate state-wise estimates. Similar studies, such as Nisbet et al. (2016), Zhang et al. (2017), and Zhao et al. (2019), also emphasize the challenges of observational density and the potential for improved accuracy with enhanced data networks.

However, we agree with the usefulness of providing region-wise emission estimates. Those have now included monthly region-wise prior and posterior emissions for 2018 and 2019 - please see Figure S13 and Figure S14 in the Supplementary Material.

Revised the Sect. 3.5 as follows:

Line 406 - 409:

"Although we focus on the national inversion estimates, owing to the inversion approach incorporated, a regional analysis (consistent with Sect. 3.1) of prior and posterior estimates has been presented in Fig. S13 and Fig. S14."

**RC:** Could the state vector also be resolved temporally to illustrate seasonal adjustments? This would be particularly interesting given the strong seasonality of emission sectors in some states

**AR:** We perform monthly inversion. State vectors are optimized on a monthly scale. Please see our response above (see Fig. (s) S13 and S14 in the revised manuscript).

**Revised the Sect. 3.5 as follows:**

Line 406 - 409:

"Although we focus on the national inversion estimates, owing to the inversion approach incorporated, a regional analysis (consistent with Sect. 3.1) of prior and posterior estimates has been presented in Fig. S13 and Fig. S14."

,,

**RC:** Moreover, the comparison of model results to TROPOMI data is limited to monthly means. A more detailed comparison is needed: how well does the model reproduce individual observations?

**AR:** For the comparison with individual TROPOMI retrievals, we emphasize that daily-scale validation is limited by both observational sparsity and random retrieval errors. TROPOMI retrievals exhibit inherent noise at individual overpass levels, particularly over cloudy regions, which limits their usefulness in model evaluation. Because of the above, the evaluation strategy considered the monthly means that reduce the random errors; thereby, we report model uncertainties at a monthly scale, which is beneficial for inversion studies.

**RC:** Much of the variability may already be explained by seasonal changes in concentrations, meaning that comparisons of total CH4 concentrations are more indicative of agreement in the CAMS product than of the model performance. What do the statistics look like for enhancements only? Summary statistics such as bias, RMSE, and correlation before and after the inversion would add substantial value.

AR: To examine the potential of our model to represent the variability from the regional fluxes compared to background (based on CAMS), we performed the following comparisons: TROPOMI vs. Enhancement (i.e. removing background contribution from total) and TROPOMI vs. Background. Figure R1 presents the results, indicating that observed variability aligns with the variability in enhancements (contributed by regional fluxes) than with the background variability. The correlation between the enhancement and observed XCH4 is consistently higher than that between the background and observed XCH4. The above analysis reconfirms the potential of our model to represent the regional variations at the monthly scale.

Figure R1: Contribution of regional and background flux fields on observed  $XCH_4$  variations, for 2018 and 2019. Correlations were calculated spatially excluding spin-up time: the boxes represent the interquartile range (25th–75th percentiles) with the median shown as a horizontal line, while the whiskers denote the minimum and maximum values of the correlation coefficient.

Additionally, for the comparison with the Thumba site, we decoupled the contributions from regional fluxes and compared the variability captured by our model.

The text is revised as follows:

**Line 319-321**

"The comparison has also been done by removing the boundary contributions from CAMS (see Fig. S8), showing that enhancements correlate ( $R^2 = 0.48$ ) with observed variability. Thus, the above comparison suggests the potential of our model in representing the regional and seasonal variations."

**RC:** Use of additional TROPOMI products: It would be highly desirable to include the other two TROPOMI XCH4 products: the operational and the GOSAT-blended product. These can differ considerably. Since the authors employ an analytical inversion (which requires only a single forward model run), incorporating these additional datasets seems feasible. This could also help better assess the posterior uncertainty, which currently appears rather low in this study.

AR: We thank the reviewer for this thoughtful suggestion. We fully acknowledge the value of multi-product assessments in improving robustness. We have carefully examined the differences among the three TROPOMI XCH4 products, including the scientific (used in our study), operational, and GOSAT-blended datasets. While there are regional differences among them, we found that these variations generally fall within the 16 ppb observational uncertainty (Figure S1). Given that the analytical inversion method used here formally incorporates observational uncertainty (16 ppb), the posterior flux adjustments, as done in the present study, may not reflect this impact, but lie within the posterior error bound. i.e. the differences between products do not exceed the uncertainty threshold built into the inversion; hence cannot be considered as a suitable setup to assess their impacts on posterior or its uncertainty.

However, we agree with the systematic observational discrepancies seen in Fig. S1 and see the importance of examining their maximum possible impact on posterior flux uncertainty in the estimations. Hence, we performed further analysis by redefining the measurement uncertainty that encompasses worst-case retrieval differences, by incorporating 50% more measurement uncertainty than originally assigned in inversion, in order to assess the maximum potential impact of these retrieval differences on posterior flux estimates at a national scale. The results of the above test are included, and the text is revised as follows:

The following text has been added in section 2.4.1:

Line: 211-220:

"The chosen measurement uncertainty encompasses the variability across TROPOMI  $XCH_4$  products ( $\leq 16$  ppb; Fig. S1). Nonetheless, as illustrated in Fig. S1, systematic differences exist among  $XCH_4$  products, which are likely to impact the optimization of fine-scale state vectors. Understanding how differences in various retrieval products influence flux estimations is vital for characterizing posterior uncertainty in inverse studies. This necessitates a more sophisticated inverse configuration that also includes a higher discretization of state space and a detailed sensitivity analysis of prior and

posterior fluxes across the region, which we consider as a future direction for this study. However, we considered the systematic observational differences due to different retrieval algorithms, as seen in Fig. S1, in a separate inversion by redefining the measurement uncertainty (>50 % more than the originally defined, ~ 25 ppb) that more-or-less represents the worst case scenario for retrieval differences across the region. The above setup can thus allow us to examine the maximum likelihood impact of such retrieval differences on posterior flux estimates at a national scale."

**Also, in section 3.5- Edited Line 408-412:**

"As explained in Sect. 2.4.1, we examined the impact of differences in the retrieval algorithm (see Fig. S1), considering the worst-case scenario, on annual posterior flux estimates for the year 2018, showing that the posterior uncertainty increased to 4.4 Tgyr-1 from 3.5 Tg yr-1 at the national scale. Future research is required for fully characterizing the impact of retrieval error uncertainty on posterior flux estimations at a fine scale by employing a more advanced inverse configuration that incorporates higher state space discretization and a detailed sensitivity analysis."

**Technical Comments**

RC: Page 2, Line 18: nearly --> more than

AR: Done (Line 20)

**RC:** Page 2, Line 28: OH- --> OH

AR: Done (Line 30)

RC: Page 2, Line 36: "largest" in the world?

**AR:** The line:

"The country holds high  $CH_4$  emission potential with its largest cattle population, intense flood irrigation practices, ever-increasing fuel demand, and large wetland extent (nearly 4.7% of its total geographical area) (Ganesan et al., 2017; Garg et al., 2011; Ministry of Environment and Change, 2015; Myhre et al., 2013b)."

**Edited to Lines 38-42:**

"The country has the largest cattle (including bovine) population (Robinson et al. (2014) in the world. Along with this, its huge intense flood irrigation practices, ever-increasing fuel demand, and large wetland extent (nearly 4.7% of its total geographical area) contribute to its high  $CH_4$  emission potential (Ganesan et al., 2017; Garg et al., 2011; Ministry of Environment and Change, 2015; Myhre et al., 2013b). "

**RC:** Page 3, Line 49: Jones et al. (2021) and Cusworth et al. (2022) are not really appropriate references at this point

**AR:** Yes. Removed and retained the other citation (Line 53).

**RC:** Page 3, Line 64: "more advanced" is disputable. It's certainly different than GOSAT.

**AR:** Edited to "more recent". (Line 69)

**RC:** Page 4, Line 74: Is there really no other in-situ site available in India to validate the simulation?

**AR:** We are unable to obtain highly precise and accurate observational records over this analysis period (two-year period) across India that can act as reference observations for model evaluation, largely due to the absence of such measurements and in part because of the individually owned data policy that hinders open access.

RC: Page 4, Line 87: "single-pass" --> "single overpass"

**AR:** Done (Line 96)

Page 5, Line 104: This likely refers to the temporal resolution of the output, not of the model itself.

**AR:** Edited to "We have used the WRF-GHG 3.9.1.1 version with a horizontal resolution of  $10 \times 10 \text{ km}^2$  (Lambert conformal conic projection grid) and an output temporal resolution of 1 hour. (Line 114-115)

**RC:** Page 5, Line 108: Does this mean that the simulated concentrations are only affected by emissions up to 18:00 the previous day? Or are the tracer concentrations (from emissions) propagated to the following day? And is also the background/CAMS tracer re-initialized every day?

**AR:** The statement in line 108 (now line 117) implies that the first 6 hours are excluded from the simulation to allow for model spin-up, specifically for meteorology. The tracer

concentrations, including those from emissions, are propagated to the following day, and the background/CAMS tracer is not re-initialized each day, but instead continues to evolve based on the initial conditions provided.

**RC:** Page 5, Line 110: Have the authors considered using a different CAMS product, such as one that is fully inversion-optimized? This might have substantially lower biases than the GQIQ product which is crucial when subtracting the background concentrations from the observations for assimilation.

AR: There is a correction in the information -it is not the GQIQ product. We have used the CAMS Global Greenhouse Gas Reanalysis (EGG4). This has been updated in Table 1. We have not used the fully inversion-optimized product. However, we have compared these two products and found that the differences remain within approximately  $\sim 13$  ppb (Figure R2). Note that the inversion-optimized product utilized SCIAMACHY satellite measurements and surface flux observations, both of which are sparse over India, the former due to cloud/aerosol issues, and the latter due to the absence of long-term observation stations over India.

**Figure R2:** Difference in CAMS EEG4 (reanalysis) and the latest version (data not available for 2018) of inversion optimised dataset (using satellite+ surface-air sampled data) for 2019.

**RC:** Page 5, Line 111: Only initial fields? Is the CAMS product not used to also provide the boundary condition of the background tracer?

**AR:** It has been used to provide the boundary condition of the background tracer as well, which has already been given in lines 118-120:

"The model is re-initialized each day with ERA5 meteorology, in which a 6-hour spin-up time was configured. For  $CH_4$  mixing ratio fields, initial and boundary conditions are

prescribed from the Copernicus Atmosphere Monitoring Service (CAMS re-analysis data)."

**RC:** Figure 1: The state boundaries are very difficult to distinguish. Since these abbreviations are used numerous times, clearer boundary delineation is important.

**AR:** Fig. 1 has been edited to show two figures in one panel: one with the regional delineation and the other with the topography height contour, to ensure clearer boundary delineation.

Table 7: The table formatting needs to be revised! Column titles must be visually separated.

**AR:** Edited

**RC:** Page 8, Line 137: "Calibration was performed periodically": How often was the calibration performed? This type of instrument is known to be temperature-sensitive, hence frequent calibration is important.

**AR:** Revised as follows, Line 149-150:

"Calibration was performed periodically. However, it should be noted that the instrument can be sensitive to temperature, requiring frequent calibration, which was not regularly met."

**RC:** Equation 1: The summation sign is missing.

**AR:** The document we submitted has the summation sign, but it is missing in the preprint. We will communicate with the journal regarding this.

Equation 9: Why does "TR" appear in the equation? Should the expression not just be " $\Phi$  perturbed –  $\Phi$ "?

**AR:** As explained earlier, TR refers to the target region that we use for constraining K-matrices.

**RC:** Page 9, Line 182: Does the optimization of anthropogenic emissions alone not lead to potential systematic errors in regions with very high wetland emissions in the northeast of the country? Have the authors considered optimizing these as well?

**AR:** Natural wetland emissions were excluded from the inverse optimization. The impact of natural wetland emissions on the column mixing ratio enhancement was minimal, smaller than the uncertainty in the satellite measurements. This has been clarified in the following lines:

Line 441-443 has been modified to:

"We excluded natural wetland emissions from the inverse optimization as they have resulted in negligible impacts on the column mixing ratio enhancement (Fig. S10), which is smaller than the uncertainty of the satellite measurement."

RC: Page 10, Line 184: "biomass emissions" --> "biomass burning emissions"?

**AR:** Done (Line 201)

**RC:** Page 10, Line 187: Are these background mixing ratios simulated by the model or derived from observations?

**AR:** Edited to (Line 204):

"Simulated background mixing ratios"

**RC:** Page 10, Line 193: Is also Sa a diagonal matrix?

**AR:** Yes. Sa is a diagonal matrix.

Edited line 210-211:

"We have not considered cross-correlations; hence, only the diagonal elements of the matrices  $S_e$  and  $S_a$  are non-zero."

**RC:** Page 10, Line 199: Do the authors mean that the emissions are indeed "spatially averaged" or are they saying that one parameter per state represents the total emissions for that state?

**AR:** Yes, we agree that "spatially averaged" is confusing here. It is rewritten as follows:

Line no: 200 - 202

"x\_A represents the prior fluxes, which consist of monthly anthropogenic (major contributions from enteric fermentation, agricultural soil, waste water handling, and fuel exploitation) and biomass burning emissions. For the optimization, we average the

emissions per state. i.e. optimizing one parameter per state representing the total emissions for that state."

**RC:** Equation 9: The summation signs are missing. Also, what unit does x have?

**AR:** Similar to the previous case, the preprint we submitted had the summation, but it is missing in the one available online. We will look into this. The annual optimised fluxes have the unit of Tgy-1. The text has been modified thus:

Line 233:

"The national budget for annual optimized fluxes  $\hat{x}_{annual}$  (in Tgy-1) is derived as: "

**RC:** Page 11, Line 211: "Regional distribution" --> "Regional and sectoral distribution"

**AR:** Done (line 238)

**RC:** Section 4.1: This section presents interesting content, but it is quite lengthy and somewhat tedious to read. A more concise and focused text would greatly enhance readability.

**AR:** The content has been made more concise.

**RC:** Page 11, Line 213: Why is EDGARv8 used here when the inversion was performed using EDGARv7?

**AR:** We used the updated version at the time.

**RC:** Figure 2: Subfigures c–f are too small to discern meaningful details in the emission maps.

**AR:** Fig.2 c-f has been moved to the supplementary section (Fig. S2) for clarity, and the text has been modified accordingly.

**RC:** Page 12, Line 240: Please add a reference to Fig. 3 when discussing the GFAS emissions.

**AR:** Done (line 267)

**RC:** Page 14, Line 260: Both links are invalid.

**AR:** On cross-checking, the links seem to be working fine in India, Europe and the US. They have been copied here for easy reference.

- 1) <a href="https://rangareddy.telangana.gov.in/animal-husbandry">https://rangareddy.telangana.gov.in/animal-husbandry</a> (line 281)
- 2) <a href="https://khammam.telangana.gov.in/economy">https://khammam.telangana.gov.in/economy</a> (line 283)

**RC:** Page 15, Line 295: Vertical mixing does not reduce column mean concentrations. Could this be due to the seasonality of OH instead? Or does this seasonality really result from the combination of increased mixing and the higher sensitivity towards near-surface concentrations of TROPOMI?

**AR:** Thank you. Edited. Line no: 345-349:

"The minimum enhancement for the whole Indian domain occurs during the monsoon season (June–September), likely due to a combination of higher boundary layer heights and stronger winds, which enhance vertical and horizontal transport affecting column  $CH_4$  concentrations. The concentrations may also be impacted by the seasonal changes in regional or larger fluxes (>1000 km); however, it needs further investigation to assess their contributions."

**RC:** Figure 4: The upper end of the color scale is too low. Seasonal patterns described in the text (p.15, line 294) are not visible in the current plot.

**AR:** Thank you for pointing this out. Figure 4 (now no. 5) is replaced with an annual mean map instead of the monthly figure panel. The monthly figure panel (now placed as Fig. S9) has been modified to show higher upper ends of the color scale and thereby represent the seasonal features in a better manner.

**RC:** Page 20, Line 339: How do the authors derive the values of 75 (72) ppb? These are not clearly traceable from the table.

**AR:** This is a typo, which has been modified in the manuscript to 13 (29) ppb. (now in line 396).

**RC:** Page 20, Line 349: The fact that the model explains 56% and 79% of the observed variability is largely due to the seasonality already captured in the CAMS product. What do these statistics look like for the enhancements only?

**AR:** A supplementary figure has been added, and it has been mentioned in the text: Line 319-321:

"The comparison has also been done by removing the boundary contributions from CAMS (see Fig. S8), showing that enhancements correlate ( $R^2$ =0.48) with observed variability. Thus, the above comparison suggests the potential of our model in representing the regional and seasonal variations."

**RC:** Page 20, Line 350: This sentence does not make sense.

**AR:** The line has been removed.

**RC:** Section 4.4: This section would make more sense if it came right after Section 4.1. This would mean that the analysis of the enhancements and the comparison with the satellite data would come before the inversion results.

**AR:** Done. It comes after section 4.1 (now 3.1)

**RC:** Figure 9: It is difficult to reconcile Fig. 7b with Fig. S7b. They do not seem consistent. In the supplementary figure, the comparison appears much better.

**AR:** We cannot expect a one-to-one compatibility, as Figure 7b (now 4b) shows the median, 25th, and 75th percentiles for both 2018 and 2019 data for each month, while Figure S7b presents the time series from January 2018 to December 2019.

**RC:** Page 21, Line 376: How do the authors arrive at the uncertainty range of 14 to 23%? Based on the total a posteriori emissions and uncertainty, I calculate a reduction of 3.8 to 26.8%.

AR: The overestimation by the EDGAR emission inventory is calculated by comparing the EDGAR-reported annual emissions for 2018 and 2019 with our derived posterior anthropogenic emissions for India. EDGAR reported 28.7 Tg emissions in both 2018 and 2019, whereas our posterior estimates are 25.2 Tg and 23.3 Tg, respectively. The range 14% - 23% is derived from this.

Uncertainty = ((Posterior - EDGAR) / Posterior) \* 100

Uncertainties for 2018 and 2019 were calculated using the above equation.

**RC:** Page 23, Line 429: The study's estimate is 24% higher than the IPCC value. This means the IPCC estimate is 19% lower than the study's result, not the other way around.

**AR:** Thank you for pointing out this confusion. Edited:

"At the same time, our estimate is 19 % higher than what the Government of India reported to the UNFCCC for the same period but close to the latest Global Methane Budget 2000-2020." (Line 488-489)

More context has been added/revised in Lines 415-429 in section 3.5:

"Previous studies also reported an overestimation of the global emission inventories over India. For instance, Qu et al. (2021) report 41–57 Tg yr-1 anthropogenic CH4 emission from India, which is significantly higher than our estimations. Also, Zhang et al. (2021) estimate Indian anthropogenic methane emissions of  $33\pm0.6$  Tg yr-1, higher than this study estimates. However, the Global Methane Budget (2000-2017, Saunois et al. (2019)), based on top-down approaches using in-situ and GOSAT observations, suggest 25 Tg yr-1 of anthropogenic CH4 emission from India, but acknowledging large uncertainty ranges in their estimations. Also, bottom-up models' estimates that are compiled in Saunois et al. (2019) and Jackson et al. (2020) indicate a mean anthropogenic CH4 emission of 21-24 Tg yr-1 from India. The above two estimates align with our results, though we used independent observations and a different modeling approach. The recent updates on the Global Methane Budget (2000-2020, Saunois et al. (2025)) indicate anthropogenic methane emissions of 37-49 Tg yr-1 for South Asia (including Afghanistan, Bangladesh, Bhutan, India, Nepal, Pakistan, and Sri Lanka), in which around 21.7 Tg yr-1 are contributed from the Indian region (calculated using the data prescribed from Martinez et al. (2024)). Janardanan et al. (2024) reported the annual averaged (2009 - 2020) CH4 emissions from anthropogenic sectors over India as  $24.2 \pm 2.1$  Tg yr-1 which is close to our results. The total CH4 emissions derived from a combination of satellite data (GOSAT), surface and aircraft measurements, and the atmospheric transport model for 2010–2015 were found to be 22 Tg yr-1, which is substantially lower than the emissions reported by the EDGAR v4.2 inventory (Ganesan et al., 2017). "

**RC:** Page 23, Line 422: Would this large effect of wetlands on XCH4 not suggest that these emissions should be optimised?

**AR**: Thank you for the feedback. There has been a modification.

Line 422 (now 479-481) has been modified to:

"The total  $XCH_4$  along the eastern coast reflects the influence of agricultural soil emissions on column-averaged methane. Although wetland emissions peak in this region, their contribution to atmospheric mixing ratios is negligible."

Given the small magnitude of the wetland contribution in our simulation setup, we chose not to explicitly optimize this component in the inversion. However, we acknowledge that optimizing wetland emissions, particularly during monsoon and post-monsoon seasons when their influence is expected to be higher, could improve model—observation agreement. This is a promising direction for future work, especially with improved wetland emission inventories or dynamic wetland models in conjunction with a good number of ground-based measurements.

---

## Author Comment (AC2)

We appreciate the editor and two anonymous referees for their time and constructive review of our manuscript. We greatly value the feedback received, have thoroughly reviewed and addressed all comments, suggestions, and concerns, and implemented necessary revisions into the manuscript. We believe that these modifications have significantly enhanced the clarity, accuracy, and overall readability of our manuscript - Thank you.

The modified texts in the revised manuscript are given in *blue italic* font.

**RC:** Referees' Comment (in black font)

**AR:** Authors' Responses (in blue regular font)

**Detailed Response to Referee #2**

**General comments**

**Summary of the study**

**RC:** This study is about estimating methane emissions from India by optimizing bottom-up emission estimates using TROPOMI data in an inverse modeling framework. In section 1 (introduction), the significance of the research question, the state of research, knowledge gaps and goals of the study are explained.

Sections 2 and 3 introduce data and methods: the datasets analyzed in this study, the atmospheric transport model, and the prior emission datasets, the sampling method and the inverse modelling framework for emission estimation.

Section 4 (results and discussion) is organized as follows. Section 4.1 describes the spatiotemporal distribution of emissions in the bottom-up datasets introduced in section 2. Section 4.2 describes the spatiotemporal distribution of XCH4 modeled using the bottom-up datasets and a boundary condition dataset. Section 4.3 and 4.4 add comparisons of the modeled mixing ratios to observed XCH4 from TROPOMI and surface CH4 from one groundbased site (Thumba), respectively. Section 4.5 shows the inverse modeling results.

The study concludes with section 5, which states the conclusions drawn from the analyses.

**Overall approach**

Overall, the study is a contribution towards an important goal, namely improving the understanding of atmospheric methane emissions from India. The methodological approach (inverse modeling of atmospheric methane transport to optimize a prior flux

estimate) is well established in the literature. That and the comparisons to similar results from the literature provide confidence that the results are sound.

**AR:** We thank the reviewer for this valuable feedback. All the comments have been addressed, and the modifications have been made accordingly in the manuscript.

**RC:** Selection of shown results**

The study puts a big emphasis on describing the spatiotemporal distribution of methane emissions from published bottom-up datasets (section 4.1, 4 pages), as well as their impact on XCH4 (section 4.2, 2 pages). In my opinion, the study abstract/introduction/conclusion) should explicitly state that such a review is one goal of the paper. Because for the goal of the study as stated in the title, "to improve methane emission assessments" using an inversion, the level of detail of these sections is too much, in my opinion. At the same time, the statements in section 4.2 and 4.3 about the separate contributions of the different sources (anthropogenic, biomass burning, wetlands) to XCH4 would actually benefit from a bit more detail, namely showing separate plots of these contributions and of the background (details in the specific comments).

**AR:** Thank you for the constructive comments. To address this, we have made the following changes in the manuscript:

We have revised the manuscript (abstract, introduction, and conclusion) to explicitly state that one of the objectives of the study is to assess the distribution of methane emissions across the study region from current bottom-up datasets and to relate them to examine spatiotemporal details of observed and simulated XCH4.

We have included separate plots showing the contributions of anthropogenic and wetland emissions to XCH, in the appropriate sections (Sect. 3.3, 3.4 & 3.5). We believe that the above revisions allow readers to infer distinct contributions of these sources to XCH, The following statement has been added to the abstract. "

**Line 6-8:**

"In addition to an inversion framework, we present a spatiotemporal assessment of bottom-up Indian methane emissions and their influence on  $XCH_4$ , supplying the context needed for regional emission optimization."

In the introduction, already in line 73-75, it is given:

"Hence, in the present study, we explore the potential of TROPOMI measurements in representing the distribution of CH4 fluxes over the Indian region along-side a spatio-seasonal analysis of the  $CH_4$  bottom-up inventory information."

**Also, we have added Line 83-86:**

"The overview of spatiotemporal analysis of methane-emission patterns across India from global bottom-up inventories, including agriculture, livestock, and fossil-fuel sectors and their impact on column-averaged methane mixing ratio enhancements, provides the context for improving or complementing current emission estimates over India."

**In conclusion: Line 468-471:**

"This study characterizes regional and seasonal methane-emission patterns from global bottom-up inventories and assesses their possible influence on  $XCH_4$  enhancements. The analysis identifies key uncertainty drivers such as the elevated anthropogenic emissions in the post-monsoon months, thereby guiding refinement of top-down  $CH_4$  estimates across India."

RC: By contrast, the section on emission estimation results - which should be the major part of a study that aims "to improve methane emission assessments" - is very short (section 4.5 - 1.5 pages): The only result given is the annual national total emissions and their uncertainty, and a comparison to the corresponding bottom-up estimate. Many questions one would expect to be addressed - and that could be addressed - using the results of the already performed inversion are not addressed. I recommend that the authors show more details of their results in section 4.5, i.e. the spatiotemporal and sectoral distribution corresponding to their state vector resolution, and whether/how much the fit to the observations (TROPOMI and Thumba) improves after the optimization. If there is a good reason why the authors wish to not show such additional details, it needs to be stated in the manuscript, and the expectations of the reader should be managed, for example by adjusting the title (e.g. by switching to a wording with "towards improving methane emission estimates").

AR: In the present study, our focus is to provide national-scale anthropogenic CH4 emission estimates, considering the coverage of the satellite observations, with the limited ground-based measurements (including column observations) in India. The inversion framework we employed in principle allows for posterior emission analysis at finer spatial and sectoral scales. However, we chose to report only the aggregated national totals, as we expect that the information content deduced from the available

assimilated observations can be insufficient to report the sectoral or sub-scale emissions accurately. As R02 rightly pointed out, the study is towards improving the national methane estimates by evaluating the feasibility of implementing WRF-GHG to utilize TROPOMI observations in the inverse framework. While a detailed exploration of sectoral and sub-regional emissions is beneficial, achieving those would demand more ground-based measurements and spatially more resolved satellite observations, which are currently limited for India. For instance, Pillai et al. (2016) highlight the limitations of satellite-based inverse modeling to derive highly resolved emissions due to insufficient observational density, suggesting that denser observation networks could enable more accurate state-wise estimates. Similar studies, such as Nisbet et al. (2016), Zhang et al. (2017), and Zhao et al. (2019), also emphasize the challenges of observational density and the potential for improved accuracy with enhanced data networks.

However, we agree with the usefulness of providing region-wise emission estimates. Those have now included monthly region-wise prior and posterior emissions for 2018 and 2019 - please see Figure S13 and Figure S14 in the Supplementary Material.

**Line 406 - 409:**

"Although we focus on the national inversion estimates, owing to the inversion approach incorporated, a regional analysis (consistent with Sect. 3.1) of prior and posterior estimates has been presented in Fig. S13 and Fig. S14."

**Also:**

We revised the manuscript title to "Leveraging TROPOMI observations and WRF-GHG modeling towards improving methane emission assessments in India" as suggested to better align with the study scope.

**RC (Language):**

A general language improvement is required. "The" is overused. Wording is often unclear or inaccurate; comparatives ("more", "growing", "enhanced", "increased") are used where positives ("large", "high") are more appropriate. Many specific comments below point to these issues; in several places I suggested alternative phrasing.

**AR**: We thank the reviewer for the careful language review. We have implemented a comprehensive language edit to improve the manuscript content. We believe these changes also improved the readability and clarity of the manuscript.

**RC** (Specific Comments):**

**Abstract**

- Line 5: Please adopt the correct definition of "WRF" throughout the document ("Weather Research and Forecasting" model)

**AR: corrected**

- Line 7: "non-negligible uncertainties" - should rather be something like "non-negligible differences", no?

**AR:** The term refers to inter-model differences due to insufficient observations with limited model refinements, possibly leading to large errors in models. Hence, we believe that "non-negligible uncertainties" fit the context (Line 9).

RC: - Line 8-9: "We find that the WRF-GHG simulations overestimate the XCH4 and underestimate the near-surface CH4 distributions".- This sentence overstates the representativeness of the single ground-based measurement site used in the analysis. It is even a point later in the conclusions that CH4 emission estimation in India would probably benefit from expanding the observation network. The authors should harmonize abstract and conclusions on this point.- Also, since this observation is so prominently placed in the abstract, I expected an explanation or at least a discussion of this observation. For example, it could hint at errors in the vertical transport, but also simply reflect the different representativeness of the columnand ground-based observations used here. Neither was investigated. Instead, the conclusion section only states (line 423-424): "Our high-resolution model is capable of capturing surface CH4 variability, especially for the well-mixed conditions, as confirmed by the ground-based CH4 observations". Thus, the significance of these differences stated here in the abstract remains unclear, and abstract and conclusions need to give the same message on that point.

**AR:** We agree. Revised abstract sentence (lines 10–11):

"We find that the WRF-GHG simulations tend to overestimate  $XCH_4$  while underestimating near-surface  $CH_4$  concentrations at the Thumba site."

Revised conclusion (lines 481-485):

"Our high-resolution model is capable of capturing surface  $CH_4$  variability, especially for the well-mixed conditions, as confirmed by the ground-based  $CH_4$

observations. However, this comparison is representative of only one station, though it is a complicated measurement location to be represented by the model owing to the influence of coastal meteorology. Such ground-based observations across India are essential for evaluating the full potential of high-resolution models in representing the atmospheric distribution of trace gases and to better constrain vertical transport processes and regional representativeness."

**RC:** - Line 9: I'm not familiar with the term "first-order inversion". So I think it should be explained in the text, replaced with a clearer term or removed (the method used is an "inversion" - I see no need for an extra qualifier like "first-order"). After reading the manuscript, I think "firstorder" refers to the fact that the state vector is rather coarse.

**AR:** Corrected- 'first-order' removed (Line 11).

**RC:** - Line 10-11: "showing that the current global emission inventories overestimate CH4 emissions considerably" - please state which inventories and by how much they overestimate the emissions.

AR: Done. Revised. Line 11-13:

"Our inversion analyses report annual  $CH_4$  emissions ranging from 23.3 to 25.2 Tg with an uncertainty of 3.3 Tg (anthropogenic sources), implying an overestimation of 14 to 20 % by the EDGAR global inventory."

**Introduction**

**RC:** - Line 19: Please cite a more recent reference about recent atmospheric greenhouse gas burdens

**AR:** Done (Line 21)

**RC:** - Line 21: If possible, please cite a more recent reference about the lifetime of atmospheric methane

**AR:** Done (Line 23-24)

RC: - Line 28-29: This statement on OH trends is not accurate. First of all, contrary to what is stated here, the cited reference Stevenson et al. 2020 concludes that OH \_increased\_ recently (from 1980 to 2014). But also, as far as I'm aware, there is no definite consensus on the OH trend. E.g. Thompson et al. 2024 see no trend and cite

studies with opposing results (https://acp.copernicus.org/articles/24/1415/2024/). There is also more context in Saunois et al. 2025 (https://doi.org/10.5194/essd-17-1873-2025) and Zhou et al. 2024 (https://doi.org/10.1088/1748-9326/ad4b47).

**AR:** Thank you for the feedback. Edited to Line 30-32

"The long-term trend in OH remains uncertain, with some studies suggesting increases (e.g. Stevenson et al. (2020)), others finding no significant trend (Thompson et al., 2024), and still others showing diverging results depending on methodology (Saunois et al., 2025)."

**RC:** - Line 50-51: Please clarify in the text what you mean by "inadequate modeling systems."

**AR:** Edited to Line 54-57:

"Previous studies over India have been limited by coarse model resolution, incomplete representation of transport processes, or lack of high-resolution emission inventories. Due to these inadequate modeling systems and sparse ground measurements, limited studies have used atmospheric  $CH_4$  observations to inform about  $CH_4$  emission flux estimates across India."

**RC:** Line 52: "imminent" is not the appropriate term here

**AR:-**Replaced "to urgent" (Line 57)

RC:- Line 70-71: "We demonstrate the advantage of operating at a high resolution comparable to TROPOMI, which may minimize forward model-related uncertainties in the carbon assimilation system over India". This sentence should be rephrased: it suggests that the authors experimented with varying transport model resolutions, which is not shown in the manuscript. The model also doesn't have that much higher resolution than others cited here (e.g. Ganesan et al. 2017 ran at 12-16 km), and the study does not address whether the forward or inverse model results shown here are superior to those of previous studies.

**AR:** We apply WRF-GHG at ~10 km resolution, comparable to TROPOMI's footprint, to enable consistent model–satellite comparisons. Here, we emphasise high density, near-daily temporal coverage and higher spatial coverage of TROPOMI measurements over India, compared to other satellite retrievals such as GOSAT that previous studies (e.g., Ganesan et al., 2017) have used. The sentence is revised for its clarity.

**Line 75-80 has been edited to:**

"We use TROPOMI XCH4 for its dense, near-daily coverage over India, enabling higher comparability with simulation of  $\sim 10$  km from our Weather Research and Forecasting model coupled with the Chemistry and Greenhouse Gas module (WRF-GHG), which may effectively minimise the forward model-related uncertainties in the carbon assimilation system over India. The performance of this high-resolution model and the advantage of using highly resolved transport fields are previously reported in Thilakan et al. 2022 and Vellalassery et al. 2021."

**RC:** "Section 4 discusses the results of the study from the data analysis conducted" – I'd remove "from the data analysis conducted"

AR: Done

**RC:** - Line 71: WRF definition (see comment above)

**AR:** Done (Line 76)

Section 2

**RC:** - Line 106: Please add the citation for ERA5 here (or cross-reference to Table 1)

AR: Done (Line 118)

**RC:** - Line 110: Provide a citation, link and version of the CAMS product used here (or crossreference to Table 1)

AR: Done (Line 122)

**RC:** - What's the model top of the WRF simulation (according to Thilakan et al. it's 50 hPa)?

Please add this to Table 1. How do you extrapolate the model profile to the top of the atmosphere to compare with TROPOMI? I.e., does your comparison method account for the CH4 decrease in the stratosphere?

**AR:** The model top pressure (50 hPa) has been added to Table 1. We expect a negligible contribution of  $CH_4$  above ~50 hPa to the total simulated  $XCH_4$ . Also, we ignored stratospheric methane chemistry in this study since the study period spans only two years. Please note that we apply the TROPOMI averaging kernel to the model profile for

minimizing the biases due to differences between the model and satellite vertical sensitivity.

This has been explained in Line no 156-164:

"The model-top is restricted to ~50 hPa (model top pressure) and the model does not take into account stratospheric  $CH_4$  chemistry, which is expected to make a negligible impact on total column averages of  $CH_4$  during the analysis time, considering the methane lifetime in the stratosphere. However, to ensure a fair comparison with observations, we applied the satellite's averaging kernel (AK), as shown in Equation 1, by considering the vertical sensitivity of the satellite instrument (Schneising et al., 2019). AK is proportional to the sensitivity profile of the measurement that is weighted with the assumed tracer profiles and provides the relation between the retrieved and known tracer profiles. i.e. Applying satellite AK to the model simulations at different vertical levels minimizes the mismatches owing to the instrument vertical sensitivities to the column observations (Eskes and Boersma, 2003; Wang et al., 2023; Schneising, 2024)."

**RC:** - Line 112: "far-field (background)": I find the term "far-field" unclear, as it can mean different things depending on context. I suggest to remove the term "far-field" and only refer to "background" or "initial and boundary conditions". Here and in Section 4.2. Similarly, the term "regional sources", used elsewhere in the manuscript, should be properly defined (along the lines of "sources within the simulated domain")

**AR:** Done. "Far-field" has been removed and replaced with "background" everywhere for consistency. In section 4.2 ( now section 3.3), 'regional sources' has already been defined as (line 338-341):

"In this section, we discuss the mixing ratio enhancements in the atmospheric column in response to spatial and temporal distributions of regional sources for the period 2018-2019. i.e. by considering only contributions from regional sources, such as anthropogenic and biomass-burning emission sources over the model domain from the emission inventories, not using CAMS-derived background  $XCH_4$  (see section 2.2)."

**RC:** - Line 123: Kretschmer time profiles: mention what does this entail - e.g. month + week day + time of day?

**AR:** Line 123 (now line 131-132) has been modified to:

"We applied temporal scaling factors to the annual EDGAR emissions using step-function time profiles and converted them to 1-hour temporal resolution, following Kretschmer et al. (2014)."

**RC:** - Section 2.3 - Thumba validation station: Add something about the area it represents or at least acknowledge that it's not representative of the whole of India. "To assess the model's performance at the surface level" is overstating the representativeness. I found something on that in Uma et al. 2024, you could briefly summarize those findings here.

**AR:** Line 143-146 has been in this section has been edited to include information about the area it represents:

"Located in southwestern India, Thumba is a tropical coastal station approximately 10 km northwest of Thiruvananthapuram and 500 m inland from the Arabian Sea. This site reflects local to regional influences, but cannot capture the full spatial variability across India."

Since Uma et al. 2024 focus on CO2, we believe that summarizing the findings doesn't suit here. However, it has already been cited (Line 148).

**RC: Section 3**

Section 3 only has one subsection, section 3.1. In my opinion, the content of section 3 belongs to the methods, i.e. I would integrate it into section 2.

**AR:** Done. Section 3 is now section 2.4 (Line 153-236)

RC: - Line 162ff: "The quantities to be optimized, represented by the state vector x with n elements x1 ,x2 ,x3 ,...,xn correspond to the total monthly emission information averaged over each political state in India." The term "total monthly emission information" appears to be inaccurate, because (1) wetland emissions were not optimized (line 182ff) and (2) the EDGAR and GFAS fluxes were optimized separately. Or not? Please adjust the text so it becomes clear what exactly was optimized.

**AR:** Edited to Line 177-179:**

"The quantities to be optimized, represented by the state vector x with n elements x1, x2, x3,...,xn correspond to the monthly, state-wise emissions from (i) anthropogenic

components (EDGAR) and (ii) the sum of anthropogenic (EDGAR) and biomass-burning (GFAS) components, both (i) and (ii) optimized separately."

**RC:** - Line 170: F(x) is not only WRF-GHG but also what is sometimes called "flux model" (in your terminology: mapping of x to phi) and here, the mapping of WRF-GHG's output to total columns (Eq. (1)). Please make this clear in the text. Suggestion: Write out the relationship between phi (emission field) and x (state vector element) as an equation. **AR:** We note that  $\varphi$  is a subset of x, representing the emission fields being optimized within the target region. Because  $\varphi$  is embedded within x, it is not possible to write a separate mapping equation. Instead, F(x) represents the forward model that maps the entire state space (including  $\varphi$ ) to total column XCH4 via forward transport and flux models. This is already explained in section 2.4.1.

The text is revised for better clarity - Line 186-187:

"where F(x) encapsulates the physics of the measurements as a function of the state vector, described here by our forward model, WRF-GHG, which includes forward transport and mapping of flux fields."

**RC:** - Line 177: "the perturbed emissions"- remove "the", I thought I had missed the introduction of the perturbation. Also, specify how you perturbed the fluxes. From the equations I infer: You constructed response functions for each state vector element by adding a small flux to the prior of the state vector element. Or was it a multiplication by a 1 + a small number?

**AR:** 'the' has been removed. Perturbed fluxes are obtained by perturbing each element of the emission flux field over the target region by small random increments.

**RC:** - Eq. (4) - Define "TR"

AR: included: TR: Target region (Line 198)

- Eq. (5) Error in the last term ("x\_A" should be "x")

**AR:** It should be  $x_A$ , as  $K(x_A)$  represents modeled concentration.

**RC:** - Line 182ff: Earlier, the authors write that the prior represented anthropogenic, wetland and biomass burning emissions. Here, it sounds like only anthropogenic and

biomass (add "burning") emissions were optimized, not wetlands. I remarked on line 162 that this it needs to be clarified what exactly was optimized separately. In addition, the authors need to state why they can omit the optimization of wetland emissions, i.e. neglect their uncertainty that was cited in the introduction.

**AR:** Clarified w.r.t previous comment. Please see Line 162 (now Line 177-179) as mentioned previously:

"The quantities to be optimized, represented by the state vector x with n elements x1, x2, x3,...,xn correspond to the monthly, state-wise emissions from (i) anthropogenic components (EDGAR) and (ii) the sum of anthropogenic (EDGAR) and biomass-burning (GFAS) components, both (i) and (ii) optimised separately."

The reason has been explained and demonstrated as a limitation in line 441 - 443 (Section 3.5):

"We excluded natural wetland emissions from the inverse optimization as we focused on major anthropogenic emission sources. Further, the natural wetland prior emissions have resulted in negligible impacts on the column mixing ratio enhancement (Fig. S10), which is smaller than the uncertainty of the satellite measurement."

We also clarify here in Line 179-180 (section 2.4.1):

"We have omitted the wetland component here since it contributed negligibly to the column-mixing-ratio enhancement."

**RC:** - Line 185: I would suggest to choose one term for fluxes inside the model domain and define it properly. Both "regional" and "local" is used in different places in the manuscript, and I wondered if different things are meant.

**AR:** Changed every 'local' to 'regional'.

**RC:** - Eq. (9): summation symbols missing

**AR:** The document we submitted has the summation sign, but it is missing in the preprint. We will communicate with the journal regarding this.

**RC:** - Line 213: Here it is stated that Edgar v8 is used, earlier it was v7 - which is it? Correct all Occurrences.

**AR:** We used the updated version at the time for the bottom-up inventory analysis.

RC: - Line 214: Remove "the" from "the emission activities"

**AR:** Done (Line 240)

RC: - Line 220: Remove "the" from "the wetland" and "the agriculture"

**AR:** Done (Line 243)

**RC:** - Line 220-221: Please clarify the sentence "The seasonal sources of CH4 include agriculture (see Fig.2 (b)) and biomass burning". I presume you mean something like "The sources with significant seasonality include agriculture and biomass burning"?

**AR:** Yes, edited accordingly (Line 244-245)

RC: - Line 222ff: "The positive anthropogenic trend in CH4 emissions in India can be expected owing to the large cattle population and agricultural activities (especially rice production and waste management) as indicated by 2010-2015 GOSAT observations (Maasakkers et al., 2019)". This sentence is confusing. Do you mean to say something like "A large anthropogenic contribution to the total CH4 emissions in India can be expected ..."? The word "trend" usually refers to multiannual trends, but is used here to describe a figure that shows one year of results. Also, a "positive trend" is not explained by "large" agricultural activities as stated here, only if they were "growing". Please clarify.

**AR:** Thank you for the clarification. We agree that "trend" is an inappropriate term here. We have revised the sentence to avoid implying growth and instead state dominance of anthropogenic sources explicitly (Line 249 - 250):

"Anthropogenic sources are expected to provide the bulk of India's  $CH_4$  emissions, especially livestock, rice cultivation, and waste management (Maasakkers et al., 2019)."

**RC:** - Line 229: Fix this reference for Edgar 4.2; in the text it's cited as "Commission et al.", and in Line 514 it shows up as "Commission, E. E. et al."

**AR:** Apologies for the erroneous citation. It has been corrected. (Line 254 & Line 628)

**RC:** Line 232: State which months are the "summer monsoon season". EDIT: Later in the manuscript, I also got confused which months are defined as winter. Please add season labels to Figures 3 and 5.

**AR:** Done. (Line 258, Figures 3 & 5 (now Fig.6) have been edited)

**RC:** Figure 1: It's hard to see the delineation of the regions (IGP, ...) because at the boundaries, only one of the lines can be seen. Please modify the figure so that the colored borders are visible for all boundaries. Maybe dashed lines work.

**AR:** Fig. 1 has been edited to show two figures in one panel: one with the regional delineation and the other with the topography height contour.

**RC:** Figure 2:

- Add the region boundaries as in Fig. 1.

Fig.2 (c) -(f) has been placed in the supplementary section (Fig. S2 now) for better clarity of all figures. Also, we acknowledge the usefulness of the region-wise boundaries in this figure (Fig. S1). However, it compromises the readability of the figure with other details. Hence, we have referred to Fig.1 (where the region delineation is clear) in the caption for better understanding.

**RC:** - Why not include a panel for the coal mining sector? The authors even discuss a hot spot from coal mining (Sect. 4.1), but it's not in any figure.

**AR:** Done. Included as S6 supplementary figure and referred to in Line 283-284 as: "Figure S6 shows the methane hotspots over the Indian domain from the coal mining sector as derived from the EDGAR inventory."

Results and discussion

Section 4.1

RC: - Line 233: "the least" -> "the minimum" (Line 250)

AR: Done (Line 259)

**RC:** - Line 233: I find the usage of percentages from here on out in this section confusing: are

they the share of monthly total fluxes? Or perhaps shares of the flux in a specific sector of the annual flux of that sector? The text seems to be about seasonality of the fluxes, but if the stated numbers are percentages of the monthly total fluxes, that seasonality is not

necessarily the same as the seasonality of the fluxes. Please clarify.

**AR:** The percentages are shares of the flux in a specific sector out of the total annual flux of that sector.

Line no: 257-259

"The analysis of monthly emissions from rice cultivation over the year 2018 indicates an increasing pattern in summer monsoon seasons, June to September (see "Agricultural soil" in Fig. 2(b)), with the maximum in August (16.9%) and September (16.5%) and the minimum in April (1.9%); percentages are shares of the annual flux."

**RC:** - Line 247 and 248: Define the regions (here: "SI" and "NI") upon first use in the text (I know they are defined in a figure caption, but as a reader, I don't want to go searching.)

**AR:** Done (Line 271 & 272)

**RC:** - Line 253: Please state that these are cities.

**AR:** Only Namakkal is a city. The rest are villages. We have mentioned that in Line 276-278:

"Four hotspots identified (see Fig. S5) in SI include one city- Namakkal (11.25°N, 78.15°E; HS1) and three villages- Mandapaka Rural (16.75°N, 81.65°E; HS2), Pasumamla (17.35°N, 78.65°E; HS3), and Mulkalappalli (18.65°N, 79.55°E; HS4)."

RC: - Lines 252ff: I can't find these hotspots on the maps. Can you mark them?

**AR:** Done in Fig S5 and referred to as mentioned previously: Line 276-278:

"Four hotspots identified (see Fig. S5) in SI include one city- Namakkal (11.25°N, 78.15°E; HS1) and three villages- Mandapaka Rural (16.75°N, 81.65°E; HS2), Pasumamla (17.35°N, 78.65°E; HS3), and Mulkalappalli (18.65°N, 79.55°E; HS4)."

**RC:** - Line 256: "has the issue of ... possibly generating more volatile gases": First, reformulate "has the issue of" in a neutral way. Second, "More volatile gases" compared to what? Do you mean "Large amounts of volatile gases"? Third, if correct please change from "volatile gases" to "methane".

**AR:** Line has been edited to:

Line no: 278- 279

" Namakkal in Tamil Nadu faces poultry waste deposition, potentially increasing methane emissions (Ramasamy and Manivel, 2019)."

**RC:** - Line 257: "increased agricultural rice emissions" - increased compared to what? I guess the authors mean "high" emissions?

**AR:** Edited to "higher agricultural rice emissions" (Line 280)

- Line 271: Define "NEI" (first usage in the text)

AR: Done (Line 290)

**RC:** - Line 272: an order "of magnitude"?

**AR:** removed. Line edited to-

Line no: 290-292

"The North East India (NEI) region shows the highest biomass burning emissions in March-May ( $\sim$ 47  $\times$  10-12 kg m2), likely due to slash-and-burn practices before planting (Deshpande et al., 2023)"

**RC:** - Lines 278-286: These considerations are the motivation for doing the present study. In my opinion, they belong in the introduction.

**AR:** Thank you for the feedback. We believe these are important points that connect the existing literature and our inference about the bottom-up estimates guiding towards the relevance of inverse optimisation. Hence, we include it in the discussion section. (Line 295-303)

**RC:** - Lines 282f: "Though inverse modeling can offer better scope in updating the CH4 budget, those approaches are limited by the insufficient coverage of mixing ratio observations and inadequate representation of the process and spatiotemporal distributions in the forward models."

- I think this sentence is missing a causal connection. And it's unclear to me what "can offer better scope" means. My suggestion would be: "Though inverse modeling can improve the CH4 budget, its ability to overcome biases in the bottom-up models used as prior estimates can be limited by insufficient coverage of mixing ratio observations".

**AR:** Edited to Line no: 299 -301:**

"Though inverse modeling can improve the  $CH_4$  budget significantly, its potential to minimize biases in the bottom-up models used as prior estimates may be limited by insufficient coverage of mixing ratio observations and its inadequate representation in the forward models."

**Section 4.2**

**RC:** - Line 289ff: Throughout this section, it's not clear to me which sources inside the model domain are considered for this XCH4 analysis: The title suggests it's only the "anthropogenic" sources, i.e. excluding biomass burning and wetlands, according to the terminology established in the methods section. However, this sentence here seems to imply that also the other sectors (biomass burning and wetlands) are included.

**AR:** Modified to:**

Line no: 338 - 341 (section 3.3)

"In this section, we discuss the mixing ratio enhancements in the atmospheric column in response to spatial and temporal distributions of regional sources for the period 2018-2019. i.e. by considering only contributions from the sum of anthropogenic and biomass-burning emission sources (mostly human-influenced in India, i.e, from agricultural residue burning and managed fires) over the model domain and not using CAMS-derived background  $XCH_4$  (see section 2.2)."

**RC:** In Fig. 4, it's yet something else: anthropogenic + biomass burning. Also, sometimes "anthropogenic + biomass burning" seems to be meant when "anthropogenic" is written (I may be wrong about that one). I get that biomass burning is at least partly from human activity in the study region, as explained a few sentences below. The text reads like biomass burning is mostly or completely anthropogenic, but that needs to be stated. Please clarify this section, e.g. by sticking to the terminology of the different sources as established in the methods section.

**AR:** We have omitted the wetland component as it contributes negligibly to the column mixing ratio enhancement as previously mentioned in Section 2.4.1.

Modified in the previous sentence.

Line no: 338-341 (section 3.3)

"In this section, we discuss the mixing ratio enhancements in the atmospheric column in response to spatial and temporal distributions of regional sources for the period

2018-2019. i.e. by considering only contributions from local sources, such as the sum of anthropogenic and biomass-burning emission sources (mostly human-influenced in India, i.e, from agricultural residue burning and managed fires) over the model domain from the emission inventories and not using CAMS-derived background  $XCH_4$  (see section 2.2)"

**RC:** - Lines 289ff again: Also, if the XCH4 from wetlands is indeed not shown in this section: it's stated in lines 327-329 that the wetlands do make a contribution to the XCH4. So this needs to be shown in a figure.

**AR:** Done. Mentioned in Line no: 341 -342:

"As mentioned in section 2.4.1, we have also omitted the wetland (biogenic) component here since it contributed negligibly to the column-mixing-ratio enhancement (see Fig. S10)."

RC: - Line 290: "biomass" -> "biomass burning"

**AR:** Done (Line 344)

**RC:** - Line 291-293: The sentence "The IGP and surrounding regions exhibit significant XCH4 enhancements (>60 ppb) from background contribution (far-field fluxes) attributed to anthropogenic and biomass-burning fluxes (see Fig. 4)" has multiple issues:

- In the previous sentence, the authors write that they discuss XCH4 enhancements due to regional sources here. Therefore, I think they meant to say "regional sources" instead of "background contribution (far-field fluxes)" here as well, otherwise the sentence doesn't seem to make sense.

**AR:** Modified accordingly, using the phrases "regional sources" (Line 343).

**RC:** - What is the significance of the number ">60ppb"? Nowhere in Fig. 4 or 5, which show the enhancements described here, does this look like a special value - it's the upper limit of the colorbar of Fig. 4, but seems to be an arbitrary cut-off. Why not state the exact numbers like "up to ..." or "from ... to ..."?

**AR:** Edited Line: 343-344:

"The IGP region exhibits significant  $XCH_4$  enhancements (from 27 ppb to 67 ppb) from regional sources attributed to anthropogenic and biomass-burning fluxes (see Fig. 5)."

Also edited in the conclusion section (line 472-473):

"The WRF-GHG simulations of XCH4 mixing ratio enhancements indicate considerable contributions from anthropogenic and biomass burning emissions, particularly in the IGP region (from 27 to 67 ppb)"

**RC:** The sentence "Seasonally, compared to summer, the highest XCH4 enhancements occur during winter (with a maximum of  $\sim 251$  ppb in January)" and Fig. 4 or 5:

- The maximum of 251 ppb is printed as "max" in Fig. 4 for January, and can presumably be seen as an outlier in Fig. 5. So I assume the two figures show the same "max" (it's not stated what the "max" is - is it snapshots as reported by WRF? Hourly averages? Something else?). However, in Fig. 5, there are even higher outliers in Oct-Nov, whereas the "max" in Fig. 4 among these two months is shown as 142.2 ppb. So it seems that Fig. 4 and 5 are inconsistent.

**AR:** 'Max' in Fig. 4 (now Fig. S9) represents the maximum value in the spatial distribution of WRF enhancement (anthropogenic+biomass burning). We agree with R02, there was an inconsistency in the texts inside figures, which has been updated (now Fig.5). The result section (section 3.3 now - line 352-358) has also been modified.

**RC:** - Line 293: "Seasonally, compared to summer" sounds confusing. I guess "compared to summer" can be deleted.

Done. (Line 344)

**RC:** - State to which region the statement refers to - I guess all of India?

**AR:** Modified. Line no. 344-345:

"Seasonally, the highest  $XCH_4$  enhancements occur during winter (with a maximum of ~ 251 ppb in January; see Fig. S9) over India."

**RC:** - The sentence "The minimum enhancement is seen in the monsoon time, possibly due to large-scale mixing in the atmosphere" has multiple issues

- What does "large-scale mixing" mean? It's not a term I'm familiar with and it's not explained in the manuscript. I guess higher wind speeds? Or do you mean higher boundary layers? Vertical transport alone doesn't affect total column measurements much, so this can't be the full story. Perhaps higher boundary layers in combination with higher

wind speeds in upper layers. Please clarify. Also, please remind the reader of the seasonality of the fluxes here.

**AR:** Thank you. Edited. Line no: 345 - 349

"The minimum enhancement for the whole Indian domain occurs during the monsoon season (June–September), likely due to a combination of higher boundary layer heights and stronger winds, which enhance vertical and horizontal transport affecting column  $CH_4$  concentrations. The concentrations may also be impacted by the seasonal changes in regional or larger fluxes (>1000 km); however, further investigation is needed to assess their contributions."

**RC:** - Specify for which region this statement is true! In Fig. 5 I can see that it's not true for South India, which is affected by monsoon. So I don't understand the given explanation.

**AR:** "whole Indian domain" phrase has been added with reference to the figure, as mentioned above.

Line no: 345 - 349:

"The minimum enhancement for the whole Indian domain occurs during the monsoon season (June–September), likely due to a combination of higher boundary layer heights and stronger winds, which enhance vertical and horizontal transport affecting column  $CH_4$  concentrations. The concentrations may also be impacted by the seasonal changes in regional or larger fluxes (>1000 km); however, further investigation is needed to assess their contributions."

**-RC:** Line 300: The phrase "high spatial distribution" doesn't make sense – please adjust.

**AR:** Modified to "high magnitude in spatial distribution" (Line 352)

**RC:** - Fig. 4:

- This figure doesn't convey much useful information: all the panels look the same, and only the numbers in the panel titles are referred to in the text (and they also show up in Fig. 5). Why not show an annual mean map here, and move the monthly panels to the supporting information.

**AR:** Done. Fig. 5 has been replaced with an annual mean map. The monthly panels are in Fig. S9. Text has also been modified accordingly

**RC:** - Fig. 5:

- No outliers are plotted for the South India region. Is that correct?

No outliers present, considering the larger interquartile range was opted for.

**Section 4.3**

**RC:** - Line 308: Do the authors mean "reasonable amounts"?

**AR:** Yes. edited accordingly. (Line 361)

**RC:** - Line 310f: Please clarify "enhanced". E.g., provide context for the term "enhanced": "enhanced values ... compared to ...". Or change to a neutral term like "highest mixing ratios"

**AR:** edited to "highest" (Line 364)

**RC:** - Line 311-313: As explained above, I think a shallower vertical distribution of emitted tracers in shallow PBLs alone should not cause large gradients in total columns from TROPOMI, as its sensitivity is rather (though not completely) uniform across the lower atmospheric levels.

**AR:** This line (365-366) has been edited to:

"These seasonal increments can be attributed to the combination of surface emissions, boundary layer height, and horizontal transport, which accumulates the effect of the distribution of tracers at lower atmospheric levels."

**RC:** - Line 313: The use of the term "increased" surface emissions is unclear to me: October-November is not the seasonal emission maximum, according to Fig. 2b. Fig. 2 does not indicate total emissions and only several components of the bottom-up inventories.

**AR:** This sentence refers to figures 3 and S3 instead, as mentioned in the edited Line 366 - 368:

"These increased regional emissions, especially from anthropogenic sources, are also seen in Fig. S3 & Fig. 3, which are typical for some parts of India like the SI and IGP during the October-November season."

RC: - Line 313f: "These increased local emissions, also seen in Fig. 3, are typical for some parts of India during the October-November season". State explicitly which parts of India and which flux components are meant here. The reference to Fig. 3 is not enough, because it's too much information to process. And in section 4.1, where seasonality is discussed, I don't see a statement about an October-November maximum either. I guess this statement might refer to biomass burning emissions in the IGP region, as mentioned two sentences later in context with aerosol load (still, state what is meant here) - however, their contribution to total emissions is tiny compared to the anthropogenic emissions, according to Fig. 3. So from the data presented, it's unclear to me whether there actually is an emission peak somewhere in this season – so it remains unclear to me what causes the large XCH4 values in Fig. 6 that is referred to here.

**AR:** There are high emissions from IGP and SI from the post-monsoon season (Refer to Fig.3 and Fig.S3). The lines 366-368 have been modified to:

"These increased regional emissions, especially from anthropogenic sources also seen in Fig. S3 & Fig. 3, which are typical for some parts of India like the SI and IGP during the October-November season."

**RC:** - Line 314: "... the likelihood of increased aerosol loads, ..." -> "the likelihood of bias in the observations due to increased aerosol loads, ..."

**AR:** Edited (Line 368-369)

**RC:** - Line 322f: "We find that the WRF simulations generally overestimate the total XCH4 mixing ratio over the Indian region compared to TROPOMI observations, peaking in winter months (>1850 ppb) (see Fig. 7)."

- The choice of highlighting the number ">1850 ppb" is unclear to me. Before, it was stated that the mean observed concentration went up to 1870 in that period, Fig. 7g states 1883.2 ppb, so why cite 1850 ppb for overestimation by the model.
- Clarify: do you mean that the WRF results peak there or the difference between model and observations?

**AR:** We acknowledge that it would be better to refer to the maximum value here. Line 377-378 has been edited to:

"We find that the WRF simulations generally overestimate the total  $XCH_4$  mixing ratio over the Indian region compared to TROPOMI observations, peaking in winter months (maximum at 1883 ppb; see Fig. 8)"

**RC:** - Line 324: I stumbled upon the phrase "suggesting the increased impact". Perhaps "suggesting a large impact" is meant.

**AR:** Edited to "large impact" (Line 379)

**RC:** -However, statements on contributions to XCH4 by sector (anthropogenic, biomass burning, wetlands) could be made much better when showing plots of all three (as suggested in my next comment).

**AR:** We believe that the newly included Fig. S10 (for wetland/biogenic components alone) and Fig. S12 (for anthropogenic components alone) serve this purpose. We have referred to them appropriately. Line 384-385:

"Wetland emissions also peak on the eastern coast, but the emissions are not found to be high enough to affect the mixing ratio enhancement considerably (see Figs. S10 & S12)."

- **RC:** Line 327-328: "Similarly, high XCH4 values observed along the eastern coast can be attributed to the wetland emissions, as seen in Fig. 2f." I have several comments about that Statement:
- In my opinion, the eastern coast doesn't stick out with particularly high observed XCH4 in Fig. 6. Perhaps in October and November, but not really in April and May. Clarify to which season/months the statement applies.-Fig. 2 shows that "agricultural soils" have, on average, larger emissions than wetlands along the eastern coast. Therefore, I expect higher contributions from this sector than from wetlands to XCH4 in that area. So the conclusion by the authors that XCH4 peaks there are explained by wetland emissions is, in my opinion, not convincing.
- If the wetland emissions had a big influence on parts of the observational dataset, it would be unclear why they were excluded from the optimization. In lines 396-398, it is even stated that the influence of wetland emissions was very small overall contradicting the statement here.

**AR:** Many thanks. We agree; it was an oversight; we analysed again and rectified the error. Line 382-383 has been modified to:

"Similarly, high  $XCH_4$  values observed along the eastern coast during October and November can be attributed to the agricultural soil emissions, as seen in Fig. S2c."

RC: - There are a few options how the authors could address this comment. They could clarify which emission dataset has the higher influence on XCH4 on the eastern coast by running them as separate tracers in the WRF model, and show wetland XCH4 contributions analogous to Fig. 4. The abovementioned lines 396-398 suggest that the authors already did run separate tracers, so adding these plots to the manuscript or supplement would not be a lot of work while adding a lot of value to this section. Or they could clarify which months they mean, and state that the "agricultural soil" dataset could fit the observed XCH4 distribution too - or explain why that is not the case.

**AR:** As the reviewer R02 suggested, we did look at the analyses. The wetland emissions can't in any way compare to the agricultural emissions in magnitude (Refer to Fig. S10). Fig. S10 shows how the biogenic (wetland) component is negligible compared to the anthropogenic component (see Fig. S12), predominated by agricultural activities as discussed in Sect. 3.1).

These lines (382-385) have been modified as:

"Similarly, high  $XCH_4$  values observed along the eastern coast during October and November can be attributed to the agricultural soil emissions, as seen in Fig. S1c. Wetland emissions also peak on the eastern coast, but the emissions are not found to be high enough to affect the mixing ratio enhancement considerably (see Figs. S10 & S12)."

**RC:** - Line 338: I feel like it's a bit of a stretch to classify October as "winter", and October - February are not classified as one season in Fig. 3. As I suggest in another comment, please add labels for the seasons to the plots.

**AR:** "winter' removed. Labels have been added.

**RC:** Figure 6 vs Fig. 7:**

- It looks as if the data coverage is different for the observations than the simulation. E.g. in February over the Tibetan plateau, it looks like there are much fewer data points in Fig. 6 than in Fig. 7. Was the data selection of model and observations identical for their comparison? Is it just that data points below the lower bound of the colorbar are shown in

white? In that case, choose a different color than white for the lower bound to distinguish from missing data.

**AR:** Thank you for pointing out the error in the figure. It was the issue with the colorbar, and henceforth, both the figures (now Fig(s) 7 & 8) have been modified.

**RC:** Section 4.4 - Line 350: This sentence is unclear: "Monthly averaged simulations and observations are found to be highly correlated in October despite their factor of magnitude differences (Figure not shown)."

**AR:** Edited to Line 313 - 314:

"In October, monthly averaged WRF-GHG simulations and TROPOMI observations are strongly correlated, although their absolute values differ substantially (Figure not shown)."

**RC:** - Line 355: Here, the performance of the model is evaluated against observations in terms of total mixing ratios. The fact that the model captured "79% of observed variability" is dominated by the seasonal cycle of the boundary conditions, so is not related to the accuracy of fluxes in the domain. It could be more informative to do this comparison after subtracting modeled boundary conditions from both datasets (although I acknowledge that the boundary conditions may have biases too).

**AR:** We have added a supplementary figure (Fig. S8 showing the comparison after subtracting modelled boundary conditions from the datasets. Line 319-321:

"The comparison has also been done by removing the boundary contributions from CAMS (see Fig. S8), showing that enhancements correlate ( $R^2 = 0.48$ ) with observed variability. Thus, the above comparison suggests the potential of our model in representing the regional and seasonal variations."

RC: - Line 360-366: Here the authors cite three possible mechanisms for the low variability and means of the mixing ratios during summer: (a) enhanced vertical mixing, (b) influx of clean air and (c) OH oxidation. The vast majority of the joint effect of (b) and (c) together could be gleaned from the contribution of a background tracer (i.e. CAMS). Hence, the suggestion in my previous comment, i.e. to separate the influence of the background, could provide insight here as well.

**AR:** The text and figure have been included as mentioned previously.

**Line 319 - 321:**

"The comparison has also been done by removing the boundary contributions from CAMS (see Fig. S8), showing that enhancements correlate ( $R^2 = 0.48$ ) with observed variability. Thus, the above comparison suggests the potential of our model in representing the regional and seasonal variations."

**Section 4.5**

**RC:** - Line 377f: "Incorporating biomass burning emissions from GFAS has an impact of +0.3 Tg yr-1 on both prior and posterior emission estimates over the Indian region". Please clarify: Does this mean that there were two simulations, one in which the GFAS emissions were held constant and one where they were optimized?

**AR:** We performed two simulations and inversions: including and excluding GFAS emissions, but keeping everything else the same. Line 399 - 401 has been modified thus: "In this section, we present estimates of India's anthropogenic CH4 budget for the period 2018-2019, derived through inverse optimization as described in Sect. 3.2. We report posterior emissions separately: one with the impact of biomass burning included and another with biomass burning excluded."

**RC:** - Line 387: Which inventory? Also EDGAR? What were the estimated emissions?

**AR:** Edited (Line 429 - 431):

"On the other hand, Raju et al. (2022) reported that the CH4 budget for peninsular India is 0.13 Tg yr-1 higher than EDGAR v6.0 inventory-based estimates for the period 2017-2018."

**RC:** - Line 388: "improve CH4 emissions in India" -> "improve CH4 emission estimation in India"

**AR:** Edited (Line 432)

**RC:** - Line 391-392. "Prior emission errors in the spatial distribution, which would have propagated from data sources or bottom-up methodology, cannot be minimized in our optimization procedures". I disagree. They were optimized, just with coarse resolution.

**AR:** Thank you for pointing this out. We agree that the unclear statement can create this confusion. The line is rephrased as follows:

Line no: 438 - 440:

"Our optimization adjusts the magnitude of prior emissions over the target region by utilizing additional information from independent measurements, but the present inverse modeling design has the limitation to minimize any flux errors in the sub-scale spatial distribution."

**RC:** - Line 393-395: This closing sentence has nothing to do with the rest of the paragraph - please reorganize.

**AR:** The line "Though GFAS inventory includes agricultural residue burning, small fires that are common in smallholders for clearing the wastes and field preparation can be missed from prior inventories, as reported in Deshpande et al. (2022)." has been reorganised to Line 452- 454:

"Another limitation could be that though the GFAS inventory includes agricultural residue burning, small fires that are common in smallholders for clearing the wastes and field preparation can be missed from prior inventories, as reported in Deshpande et al. (2022)."

**RC:** - Lines 396ff: In my opinion, the statements on the various emission estimates in the literature belong in the introduction.

**AR:** Here, we discuss the various emission estimates directly relevant to interpreting our results and diagnosing limitations. Hence, we have included it in this section.

**RC:** - Line 396-398: "We excluded natural wetland emissions from the inverse optimization as we focused on major anthropogenic emission sources. Further, the natural wetland prior

emissions have resulted in negligible impacts on the column mixing ratio enhancement (Figure not shown), which is smaller than the uncertainty of the satellite measurement." I would remove the first sentence - it's not a justification why excluding wetlands from the optimization would be reasonable in this case. Also, as I suggested above, it would add a lot of value to show this figure.

**AR:** Please see our responses w.r.t similar comments above. Please see our revisions also.

Line 441 - 443 has been modified to

"We excluded natural wetland emissions from the inverse optimization as they have resulted in negligible impacts on the column mixing ratio enhancement (Fig. S10), which is smaller than the uncertainty of the satellite measurement."

**RC:** - Finally, the statement contradicts the above discussion of how wetlands allegedly have a noticeable influence on observed XCH4 at the eastern coast (see my comment on that above).

These lines (382-385) have been modified to imply agricultural soil emissions rather than wetland emissions as mentioned previously:

"Similarly, high  $XCH_4$  values observed along the eastern coast during October and November can be attributed to the agricultural soil emissions, as seen in Fig. S1c. Wetland emissions also peak on the eastern coast, but the emissions are not found to be high enough to affect the mixing ratio enhancement considerably (see Figs. S10 & S12)."

**RC:** - Line 399: "However, Indian wetland emissions also vary based on inventories and prior models" - unclear. Do you mean they "vary among bottom-up models"?

AR: Edited to "vary among bottom-up models" (Line 443)

- Line 399: "Approximately 7.5 Tg CH4 emissions from the Indian region ..." -> "Approximately 7.5 Tg CH4 \_wetland\_ emissions from the Indian region ..."

**AR:** Edited to "7.5 Tg wetland CH4 emissions" (Line 444)

**Section 5 (Conclusions)**

RC: - Line 418-419: "The highest seasonal enhancements of anthropogenic XCH4 occur during winter, influenced by agricultural emissions, biomass burning, and atmospheric winter transport". As stated above, I think the authors did not sufficiently demonstrate how exactly winter transport (i.e., shallow PBL height) influenced the total columns of CH4. I think they could do so by calculating mean wind speeds in the atmospheric boundary layer, which are the winds that the emitted methane is transported with, and may find that they are higher in summer presumably because the boundary layer reaches higher. Or cite a study where this was shown

AR: Line 475 - 476 has been added as:

"This inference aligns with previous studies (eg. Patra et al. (2011)) that show stronger vertical mixing during the summer, associated with higher boundary layers and faster wind speeds, may impact  $CH_4$  columns."

**RC:** - Line 422-423: "The total XCH4 along the eastern coast suggests the effects of wetlands on atmospheric column mixing ratios." As stated above, I am not convinced that it isn't rather the agricultural soils. As also remarked above, this statement contradicts the statement elsewhere in the manuscript that wetland emissions had a "negligible" effect on XCH4

**AR:** Thank you for the feedback. There has been a modification. Line no. 479 - 481:

"The total XCH4 along the eastern coast reflects the influence of agricultural soil emissions on column-averaged methane. Although wetland emissions peak in this region, their contribution to atmospheric mixing ratios is negligible."

**RC:** References - Many links follow the pattern "https://doi.org/https://doi.org/". While they seem to work, this is probably not intended.

**AR:** Edited